# Loss Dynamics of Temporal Difference Reinforcement Learning

**Blake Bordelon, Paul Masset, Henry Kuo & Cengiz Pehlevan**
John Paulson School of Engineering and Applied Sciences,
Center for Brain Science,
Kempner Institute for the Study of Natural & Artificial Intelligence,
Harvard University
Cambridge MA, 02138
`blake_bordelon@g.harvard.edu, cpehlevan@g.harvard.edu`

## Abstract

Reinforcement learning has been successful across several applications in which agents have to learn to act in environments with sparse feedback. However, despite this empirical success there is still a lack of theoretical understanding of how the parameters of reinforcement learning models and the features used to represent states interact to control the dynamics of learning. In this work, we use concepts from statistical physics, to study the typical case learning curves for temporal difference learning of a value function with linear function approximators. Our theory is derived under a Gaussian equivalence hypothesis where averages over the random trajectories are replaced with temporally correlated Gaussian feature averages and we validate our assumptions on small scale Markov Decision Processes. We find that the stochastic semi-gradient noise due to subsampling the space of possible episodes leads to significant plateaus in the value error, unlike in traditional gradient descent dynamics. We study how learning dynamics and plateaus depend on feature structure, learning rate, discount factor, and reward function. We then analyze how strategies like learning rate annealing and reward shaping can favorably alter learning dynamics and plateaus. To conclude, our work introduces new tools to open a new direction towards developing a theory of learning dynamics in reinforcement learning.

## 1 Introduction

Reinforcement learning (RL) is a general paradigm which allows agents to learn from experience the relative value of states in their environment and to take actions that maximize long term rewards [1]. RL algorithms have been successfully applied in a number of real world scenarios such as strategic games like backgammon and Go, autonomous vehicles, and fine tuning language models [2–7].

Despite these empirical successes, a theoretical understanding of the learning dynamics and inductive biases of RL algorithms is currently lacking [8]. A large fraction of the theoretical work has focused on proving convergence and deriving bounds both in the asymptotic [9–14] and non-asymptotic [15–17] limits, but do not provide a full picture of the evolution of the learning dynamics.

A desired feature of a candidate theory is to characterize the influence of function approximation to RL dynamics and its performance. Early versions of RL operated in a tabular setting, similar to dynamic programming [18], where all the states in the environment could be mapped one-to-one to a specific value and policy. In large and complex environments, it is not possible to enumerate all the states in the environment necessitating the use of function approximation for the target value and policy functions. Indeed, the recent success of many RL algorithms relies on deep reinforcement

37th Conference on Neural Information Processing Systems (NeurIPS 2023).

learning architectures that combine an RL architecture with deep neural networks to build effective value estimators and policy networks [19].

One difficulty in analysing these algorithms compared to supervised learning settings is that the distribution of the data received at each time-step is not stationary. This non-stationarity arises from two principal sources: First, whether in an episodic or continuous setting, states visited within a learning trajectory are dependent on the recent past. Trajectories might be randomly sampled but points within a trajectory are correlated. Second, when the policy is updated it also changes the distribution of future visited states.

Here, we will focus on the first form of non-stationarity when learning a value function in the context of *policy evaluation* [1] using a classical RL algorithm, temporal difference (TD) learning [20]. We develop a theory of learning dynamics for RL in this setting in a high dimensional asymptotic limit with a focus on understanding the role of linear function approximation from a set of nonlinear and static features. In particular, we leverage ideas from recent work in application of statistical physics to machine learning theory to perform an average over the possible sequences of features encountered during learning. Our contributions are as follows:

- We introduce concepts from statistical physics, including a path integral approach to describe dynamics [21–25] and the Gaussian equivalence assumption [26–29], to derive a theory of learning dynamics in TD learning (§3) in an online setting. We provide an analytical formula for the typical case learning curve for TD learning.

- We show that our theory predicts scaling of the learning convergence speed and performance plateaus with parameters of the problem including task-feature alignment [30], learning rate, discount factor or batch size (§4 and §5). Task-feature alignment is a metric that quantifies how features allow fast or slow learning for a given task.

- We show our theory can be used to understand and guide design principles when choosing meta-parameters. Specifically, we show that we can use our theory to infer optimal schedules of learning rate annealing and the effects of reward shaping (§5 and §6).

## 2 Problem Setup and Related Works

### 2.1 Problem Setup

We consider a set of states denoted by $s$, possibly continuous, and a fixed policy $\pi$ which generates a distribution over actions given the state. The state dynamics are defined by a distribution $p(\tau)$ over trajectories through state space $\tau = \{s_1, s_2, ..., s_T\}$. Note that state transitions do not have to be Markovian, but each trajectory is i.i.d. sampled from $p(\tau)$. We consider trajectories of length $T$. Each state is represented by an $N$-dimensional feature vector $\psi(s) \in \mathbb{R}^N$, so that trajectory generates a collection of feature vectors $\{\psi(s_t)\}_{t=1}^T$. The rewards are generated by a reward function $R(s)$ which depends on the state. (In general, the features and rewards can depend on action as well: transition dynamics are still fixed as the policy is fixed, but variance over rewards at a given state may need to be modeled, see Appendix B.5).

At any time, we are interested in characterizing the *value function* associated with a state, which measures the expected discounted sum of future rewards when starting in state $s_0$

$$V(s_0) = R(s_0) + \sum_{t \geq 1} \mathbb{E}_{s_t|s_0} \gamma^t R(s_t) = R(s_0) + \gamma \mathbb{E}_{s_1|s_0} V(s_1). \tag{1}$$

We use linear function approximation to learn the value function $\hat{V}(s) = \psi(s) \cdot \boldsymbol{w}$. Similar to kernel learning [31], the features $\psi$ should be high dimensional so that they can express a large set of possible value functions.

We study TD learning dynamics given this setup. At each step of the TD iteration, we sample a batch of $B$ independent trajectories from the distribution and compute the TD update

$$\boldsymbol{w}_{n+1} = \boldsymbol{w}_n + \frac{\eta_n}{TB} \sum_{\mu=1}^{B} \sum_{t=1}^{T} \Delta_n^\mu(t) \psi(s_n^\mu(t)),$$

$$\Delta_n^\mu(t) \equiv R(s_n^\mu(t)) + \gamma \hat{V}(s_n^\mu(t+1)) - \hat{V}(s_n^\mu(t)). \tag{2}$$

We operate in a online batch regime as the trajectories in each batch are resampled at each iteration. This is distinct from an offline setting where the batches would be resampled from a finite-sized buffer [1]. Convergence considerations for infinite-batch online TD learning width different types of features $\psi$ are outlined in Appendix A. The specific form for the TD-error $\Delta_n^\mu(t)$ depends on the precise variant of TD learning that is used. Here, we will focus on TD(0) but our approach can be extended to other TD learning rules and definitions of the return function. We see that the iterates $w_n$ will form a stochastic process as each sequence of states in an episode $\{s_n^\mu(t)\}$ are drawn randomly from $p(\tau)$. In general, we allow the learning rate $\eta_n$ to depend on iteration, an important point we will revisit later. The distribution of features $\{\psi(s_n^\mu(t))\}$ over random trajectories $\tau$ is in general quite complicated, depending on the details of the state transitions and the nonlinear feature maps, which motivates the following question:

**Question:** *How can the stochastic dynamics of temporal difference learning be characterized for complicated trajectory distributions $p(\tau)$ and feature maps $\psi(s)$?*

To address this question, in this work, we provide an analysis of TD learning that explicitly models the statistics of stochastic semi-gradient updates to $w_n$. Our framework is based on a Gaussian equivalence ansatz for TD learning and high dimensional mean field theory which predicts the statistics of TD errors $\Delta_n^\mu(t)$ and the weight iterates $w_n$. The theory reveals a rich set of phenomena including plateaus unique to SGD noise in TD learning which can be ameliorated with learning rate annealing.

## 2.2 Related Works

The dynamics of TD learning have been notoriously difficult to analyse. Unlike supervised learning settings, sampled states are correlated across a trajectory and the algorithms involve bootstrapping: using estimates of the value function for future states in the temporal difference update [1]. Some prior works study the least-square TD learning rule, which solves, at each step $n$ of the algorithm, a linear system for the instantaneous best fit to $n$ samples [32–34]. Alternatively, many works focus on the online SGD version of TD learning, where incremental updates are made to the parameters at each step, using fresh samples. This is the setting of our work. The focus of this literature has initially been to prove convergence and bounds on asymptotic behavior [11–14, 35]. More recently, progress has been made in deriving bounds in the non-asymptotic regime. Initial work assumed that data samples were *i.i.d.* [15–17, 36] and recent work has extended those approaches to Markovian noise [15, 37–39]. The majority of these proofs use the ODE-like method for stochastic approximation [11, 40], which corresponds to a limit of the stochastic semi-gradient dynamics where the effects of mini-batch noise are neglected. This is also known as the "mean-path" dynamics of TD learning and will correspond to the infinite batch limit of our theory. Furthermore, many of these methods require the use of iterative averaging of the learned value function, whereas we study the final iterate convergence. The approach we take here differs from many of these results as our goal is not to provide bounds on worst-case behavior but instead to provide a full description of the dynamics of the typical case scenario during learning.

Our approach also highlights the importance of the structure of the representations in controlling the dynamics of learning. This had been long been recognized in reinforcement learning and previous works proposed to improve feature representations to improve algorithmic performance [41–43]. This line of work has shown the importance of the relative smoothness of the representations and target functions in the ODE limit of TD dynamics [43, 44]. Similarly, several methods have been proposed to empirically learn a better shaping function [45, 46]. In *policy learning* it has also been recognized that using a gradient aligned to the statistics of the tasks, such as the natural gradient [47] can greatly speed up convergence [48]. Our work does not explore such feature learning per se but could be used as a diagnostic tool to analyse how representations impact learning speed.

We adopt the perspective of statistical physics, by working with a simplified feature distribution which captures the learning dynamics and solving the theory in a high-dimensional limit [49–51]. We derive TD reinforcement learning curves from a mean field theory formalism which is exact for infinite dimensional features and batch size. Similar calculations for supervised learning on Gaussian data have been shown to provide an accurate description of high dimensional dynamics [52–54]. Further, even when data is not actually Gaussian, several algorithms, such as kernel or random-features regression, exhibit universality in their loss behavior, enabling analysis of the learning curve with a simpler Gaussian proxy [26–28, 30, 55]. We exploit this idea in the TD learning setting to some success. We note that Gaussian equivalence or universality is not a panacea, and in many cases the Gaussian proxy can fail to capture important machine learning phenomena [27, 56, 57].

# 3 Theoretical Results for Online TD Learning

## 3.1 Computation of Learning Curves

We develop a dynamical mean field theory (DMFT) formalism can be utilized to compute the learning curves. We provide the full derivation of the DMFT in Appendix B. This computation consists of tracking the moment generating function for the iterates $\boldsymbol{w}_n$ over the trajectories of randomly sampled features $\{\boldsymbol{\psi}_\mu^n(t)\}_{t=1}^T$. In an appropriate high dimensional asymptotic limit, the results of our theory can be summarized as the following proposition.

**Proposition 3.1.** *Let $N, B \to \infty$ with $B/N = \mathcal{O}(1)$ and episode length $T = \mathcal{O}(1)$. Let the ground truth reward function be $R(s) = \boldsymbol{w}_R \cdot \boldsymbol{\psi}(s)$ and value function $V(s) = \boldsymbol{w}_{TD} \cdot \boldsymbol{\psi}(s)$ in the basis of our features. Define matrices*

$$\bar{\boldsymbol{\Sigma}} \equiv \frac{1}{T} \sum_t \boldsymbol{\Sigma}(t,t), \quad \bar{\boldsymbol{\Sigma}}_+ \equiv \frac{1}{T} \sum_t \boldsymbol{\Sigma}(t, t+1), \quad \boldsymbol{A} \equiv \bar{\boldsymbol{\Sigma}} - \gamma \bar{\boldsymbol{\Sigma}}_+, \tag{3}$$

*and assume that the features are such that matrix $\boldsymbol{A}$ is of extensive rank in $N$. Then the typical value estimation error $\mathcal{L}_n = \left\langle \left( V(s) - \hat{V}_n(s) \right)^2 \right\rangle_s$ after $n$ steps has the form*

$$\mathcal{L}_n = \frac{1}{N} Tr \bar{\boldsymbol{\Sigma}} \boldsymbol{M}_n, \tag{4}$$

$$\boldsymbol{M}_{n+1} = (\boldsymbol{I} - \eta \boldsymbol{A}) \boldsymbol{M}_n (\boldsymbol{I} - \eta \boldsymbol{A})^\top + \frac{\eta^2}{\alpha^2 T^2} \sum_{tt'} Q_n(t,t') \boldsymbol{\Sigma}(t,t') \tag{5}$$

$$Q_n(t,t') = \frac{1}{N} \left\langle (\boldsymbol{w}_R - \boldsymbol{w}_n)^\top \boldsymbol{\Sigma}(t,t')(\boldsymbol{w}_R - \boldsymbol{w}_n) \right\rangle + \frac{\gamma}{N} \left\langle (\boldsymbol{w}_R - \boldsymbol{w}_n)^\top \boldsymbol{\Sigma}(t, t'+1) \boldsymbol{w}_n \right\rangle$$

$$+ \frac{\gamma}{N} \left\langle \boldsymbol{w}_n^\top \boldsymbol{\Sigma}(t+1, t')(\boldsymbol{w}_R - \boldsymbol{w}_n) \right\rangle + \frac{\gamma^2}{N} \left\langle \boldsymbol{w}_n^\top \boldsymbol{\Sigma}(t+1, t'+1) \boldsymbol{w}_n \right\rangle, \tag{6}$$

*where $\alpha = B/N$ and $Q_n(t,t') = \langle \Delta_n(t) \Delta_n(t') \rangle$ is the correlation of randomly sampled TD-errors at episodic times $t, t'$ and iteration $n$. The average over weights $\langle \rangle$ denotes a Gaussian average whose moments are related to $\boldsymbol{M}_n$. The correlation function $Q_n(t,t')$ depends on $\boldsymbol{M}_n$ and the average weights $\langle \boldsymbol{w}_n \rangle$; we provide its full formula in Appendix B.3, equation (B.17).*

*Proof.* The full derivation is in Appendix B. At a high level, we track the moment generating function of the iterates $\boldsymbol{w}_n$ over random draws of features $\{\boldsymbol{\psi}_n^\mu(t)\}$, $Z[\{\boldsymbol{j}_n\}] = \mathbb{E}_{\{\boldsymbol{\psi}_n^\mu(t)\}} \exp\left( i \sum_n \boldsymbol{j}_n \cdot \boldsymbol{w}_n \right) \propto \int \mathcal{D}q \exp\left( \frac{N}{2} S[q, \{\boldsymbol{j}_n\}] \right)$ where $S$ is a $\mathcal{O}(1)$ action and $q$ are a set of order parameters of the theory which include the following overlaps $C_n(t,t') = \frac{1}{N} \boldsymbol{w}_n^\top \boldsymbol{\Sigma}(t,t') \boldsymbol{w}_n$ and $Q_n(t,t') = \frac{1}{B} \sum_{\mu=1}^B \Delta_n^\mu(t) \Delta_n^\mu(t')$. In this high dimension $N, B \to \infty$ limit with $B/N = \mathcal{O}(1)$ and episode length $T = \mathcal{O}(1)$, the order parameters can be obtained from saddle point integration, which requires solving $\frac{\partial S}{\partial q} = 0$. This procedure results in a deterministic learning curve given in equations (4),(5),(6) even though the realization of sampled states are disordered. The TD-error variables $\Delta_n(t)$ become mean zero Gaussians and the $\{\boldsymbol{w}_n\}$ also follow a Gaussian distribution with mean and variance determined by the order parameters. $\square$

Before we explore the predictions of this theory, we first make a few remarks about this result.

*Remark* 1. Though the theory is technically derived for large batch size $B$, we will show that it provides an accurate description of the loss trajectory even for batches as small as $B = 1$. An alternative formulation in terms of recursive averaging reveals transparently which approximations lead to the same result as the mean field theory (Appendix B.6).

*Remark* 2. The case where the reward function and/or the value function are inexpressible by the features $\psi$ can also be handled within this framework. In this case, the unlearnable components of the value function act as additional noise which limits performance [29]. These can also be handled by our theory, see Appendix A.

*Remark* 3. The limit where $\gamma = 0$ recovers known results in online supervised learning with stochastic gradient methods [29, 58, 59]. In this limit, the dynamics will converge to zero loss provided the model features are sufficiently rich to represent the true value function.

*Remark* 4. The TD learner with perfect coverage (infinite batch size) at each step will converge to the ground truth $\boldsymbol{w}_{TD} = \left(\bar{\boldsymbol{\Sigma}} - \gamma\bar{\boldsymbol{\Sigma}}_+\right)^{-1}\bar{\boldsymbol{\Sigma}}\boldsymbol{w}_R$ (see Appendix A).

*Remark* 5. $\boldsymbol{M}_n$ is equivalently defined as $\boldsymbol{M}_n = \left\langle (\boldsymbol{w} - \boldsymbol{w}_{TD})(\boldsymbol{w} - \boldsymbol{w}_{TD})^\top \right\rangle_{\{\tau_{n'}^\mu\}_{n'<n}}$, which measures deviation from the fixed point of gradient flow (vanishing learning rate) dynamics $\boldsymbol{w}_{TD}$ over random sets of sampled episodes (Appendix B).

## 3.2 Gaussian Approximation

The theory presented in Section 3.1 relies on an approximation of the feature distribution as Gaussian. Similar approximations have been successfully utilized in high dimensional regression problems even when the true features are non-Gaussian [26–29]. We note that an exact, non-asymptotic theory for non-Gaussian features can be provided which closes under knowledge of the fourth cumulants of the features as we show in Appendix D, though this theory is especially cumbersome to analyze or evaluate compared to the theory of Section 3.1. Concretely, Proposition 3.1 relies on the following.

**Gaussian Feature Assumption.** *The learning curves for a TD learner with high dimensional features $\{\boldsymbol{\psi}(s_t)\}_{t=1}^T$ over random $\tau$ are well approximated by the learning curves of a TD learner trained with Gaussian features $\boldsymbol{\psi}_G \sim \mathcal{N}(\boldsymbol{\mu}, \boldsymbol{\Sigma} + \boldsymbol{\mu}\boldsymbol{\mu}^\top)$ with matching mean and correlations*

$$\boldsymbol{\mu}(t) = \langle \boldsymbol{\psi}(s_t) \rangle_{\tau \sim p(\tau)} \ , \quad \boldsymbol{\Sigma}(t,t') = \left\langle \boldsymbol{\psi}(s_t)\boldsymbol{\psi}(s_{t'})^\top \right\rangle_{\tau \sim p(\tau)} . \tag{7}$$

*where averages are taken over sequences of states $\{s(t)\} \sim p(\tau)$.*

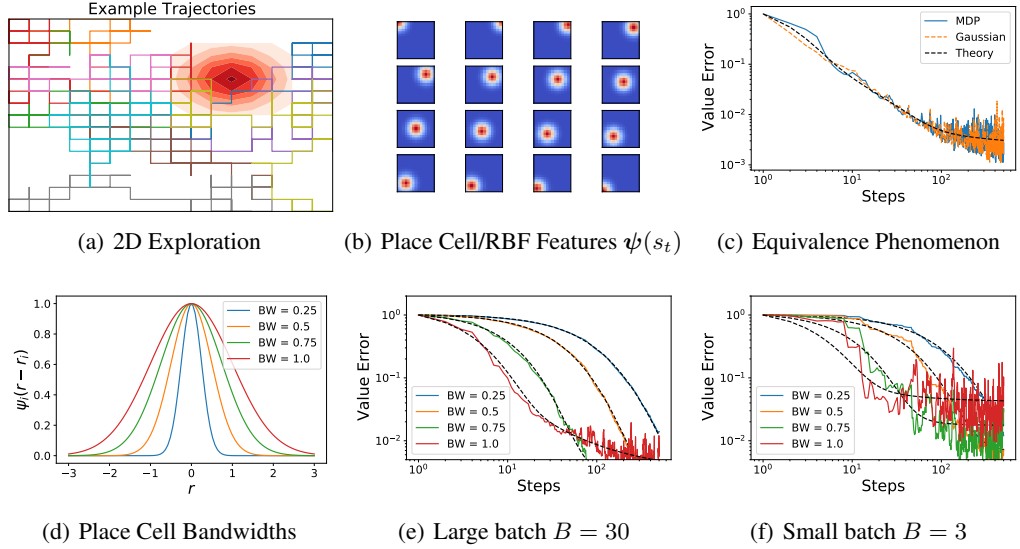

(a) 2D Exploration     (b) Place Cell/RBF Features $\boldsymbol{\psi}(s_t)$     (c) Equivalence Phenomenon

(d) Place Cell Bandwidths     (e) Large batch $B = 30$     (f) Small batch $B = 3$

Figure 1: An illustration of our theory for TD learning. (a) A diffusion process in a 2D grid world generates many possible trajectories through state space. Each colored line is a different trajectory. Reward function is shown in red, with darker red indicating higher reward. (b) When combined with nonlinear place cell feature representation, the state transitions generate a distribution over observed features $\{\boldsymbol{\psi}(s_t)\}$. (c) The value error associated with TD learning for a bump reward function on the true features generated from a single set of MDP trajectories (blue) is compared to training on sampled Gaussian vectors $\{\boldsymbol{\psi}_t\}$ with matching within-episode covariance structure. These single runs of TD learning on either set of features are consistent with the typical case theory (black dashed). (d) The structure of the features alters learning dynamics. We consider, for simplicity, altering the bandwidth (BW) of the place cell features. (e) Varying place cell BW changes the dynamics for both large batch ($B = 30$) and (f) small batch ($B = 3$) TD learning. There is an optimal BW for a given step size. Small batch stochastic semi-gradient noise is more severe.

One interpretation of this ansatz is that the dependence of the learning curve on higher order cumulants of the features is negligible in high dimensional feature spaces under the square loss. This

approximation has been shown to provide an accurate description on realistic supervised learning settings with non-Gaussian data with the square loss in prior works [26, 27, 29, 30, 55, 58]. As shown in these works, for standard supervised learning, even highly non-Gaussian features $\{\boldsymbol{\psi}(s_t)\}$ have least squares learning curves which are only sensitive to the first two cumulants of the distribution. We do not aim to provide a rigorous proof of this ansatz for TD learning but instead compute the learning curve implied by this assumption and compare to experiments on simple Markov Decision Processes (MDPs). The benefit of this hypothesis in the RL setting is that it abstracts away details of transitions in the state space and instead deals with the correlations of sampled features through time.

To illustrate an example of the Gaussian Equivalence idea, in Figure 1, we consider an MDP which is defined by diffusion through a 2-dimensional (2D) state space (Figure 1(a)). We choose the features $\boldsymbol{\psi}(s)$ to be a collection of localized 2D Radial Basis Function (RBF) bumps which tile the 2D space, similarly to the "place cell" neurons found in the mammalian hippocampus [60, 61] (Figure 1(b)). The feature map is parameterized by the bandwidth of individual "place cells". In Figure 1(c), we show the value error learning curve as a function of the number of steps $n$ (blue) and compare the value estimation error of the MDP with a Gaussian distribution for $\boldsymbol{\psi}(t)$ with matching first and second moments (orange). Lastly, we plot the theoretical prediction of our theory (described in Section 3), which is computed under the Gaussian equivalence ansatz (black dashed). We see a remarkable match of the three curves. The equivalence can be used to predict the speed of TD learning for different features, such as place cells with varying bandwidth as we illustrated in Figure 1 (d)-(f). In Figure 1 (e) and (f), we plot the loss trajectories for a single run of TD for each feature set. We observe that bandwidth affects both the learning dynamics and the asymptotic error with an optimal bandwidth at any step. One of our goals will be to elucidate the role of feature quality in learning dynamics. While the large batch dynamics are approximately self-averaging, as shown by the fact that single runs of TD learning coincide with our theoretical typical case theory curves, there is significant semi-gradient variance in the value error at small batch sizes. While we expect Gaussian equivalence to hold for high dimensional features, in low dimensions non-Gaussian effects can significantly alter the learning curves as we show in Appendix D.1. However, for high dimensional features, the equivalence holds for many other feature distributions such as polynomial and fourier features (Appendix E).

# 4    Spectral Perspective on Hard Reward Functions

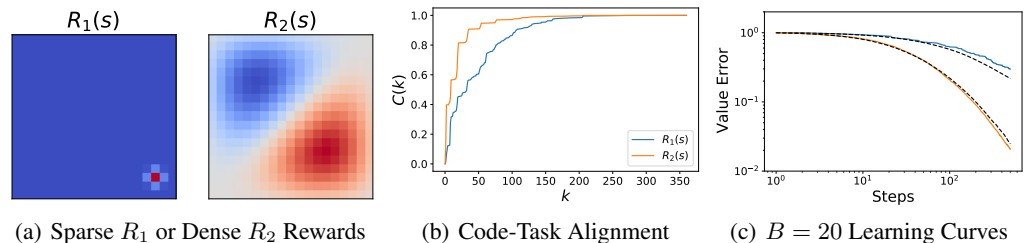

(a) Sparse $R_1$ or Dense $R_2$ Rewards    (b) Code-Task Alignment    (c) $B = 20$ Learning Curves

Figure 2: Reward functions and dynamics which lead to value functions with high spectral alignment to the features can be learned more quickly than those that do not. (a) A sparse and dense reward function in a 2D spatial navigation task can illustrate this effect. (b) The cumulative power distribution $C(k)$ defined from the spectral decomposition of $\boldsymbol{A} = \bar{\boldsymbol{\Sigma}} - \gamma\bar{\boldsymbol{\Sigma}}_+$. Concretely we let $\boldsymbol{A}\boldsymbol{u}_k = \lambda_k\boldsymbol{u}_k$ with $\lambda_k$ ordered by real part and $\boldsymbol{w}_{TD} = \sum_k w_k\boldsymbol{u}_k$. In the $B \to \infty$ limit the task which has rapidly rising $C(k) = \frac{\sum_{\ell < k} w_\ell^2}{\sum_\ell w_\ell^2}$ will converge more quickly than the task with slowly rising $C(k)$. (c) Indeed, for large batch regime ($B = 20$) the value error decreases more rapidly for $R_2$ than for $R_1$.

Our theory can provide some insights into the structure of tasks which can be learned easily and which require more sampled trajectories to estimate based on spectral decompositions of the feature covariances. We note that similar spectral arguments have been given in the ODE-limit [44] and are intimately related to the source conditions used in recent work to identify power-law rates in the large batch regime [39].

To build our argument, we diagonalize the matrix $\boldsymbol{A} = \bar{\boldsymbol{\Sigma}} - \gamma\bar{\boldsymbol{\Sigma}}_+$, obtaining $\boldsymbol{A}\boldsymbol{u}_k = \lambda_k\boldsymbol{u}_k$, noting that eigenvalues $\lambda_k$ can be complex. We then expand the TD solution in this basis $\boldsymbol{w}_{TD} = \sum_k w_k\boldsymbol{u}_k$.

The theory predicts that, the average learned weights will be $\langle \boldsymbol{w}_n \rangle = \sum_k |1 - \eta\lambda_k|^n e^{i\theta_k n} w_k \boldsymbol{u}_k$, where $|\cdot|$ is complex modulus and $\theta_k = \text{Arg}(1 - \eta\lambda_k)$. We can therefore order the modes by their convergence timescales $|1 - \eta\lambda_k|$. Given this ordering of timescales, we can order the modes $k$ from those with smallest to largest timescales. Given this ordering, we see that tasks can be learned efficiently are those with most of the norm of $\boldsymbol{w}_k$ in the modes with small timescales. We quantify how well aligned a task is to a given feature representation by computing a cumulative power distribution for the target weights $C(k) = \frac{\sum_{\ell < k} w_\ell^2}{\sum_\ell w_\ell^2}$. If this quantity rises rapidly with $k$ then the task can be learned from a small number of samples [30].

We consider again, the setting of Figure 1, the 2D exploration MDP but now contrast two different reward functions. In Figure 2 we show that this spectral decomposition can account for the gaps in loss for a place cell code in learning a sparse or dense reward function (Figure 2(a)). As expected the cumulative power rises more rapidly for the dense reward function $R_2(s)$ (Figure 2(b)). As a consequence, the value error converges to zero more rapidly than for the sparse rewards.

# 5 Stochastic Semi-Gradient Learning Plateaus and Annealing Strategies

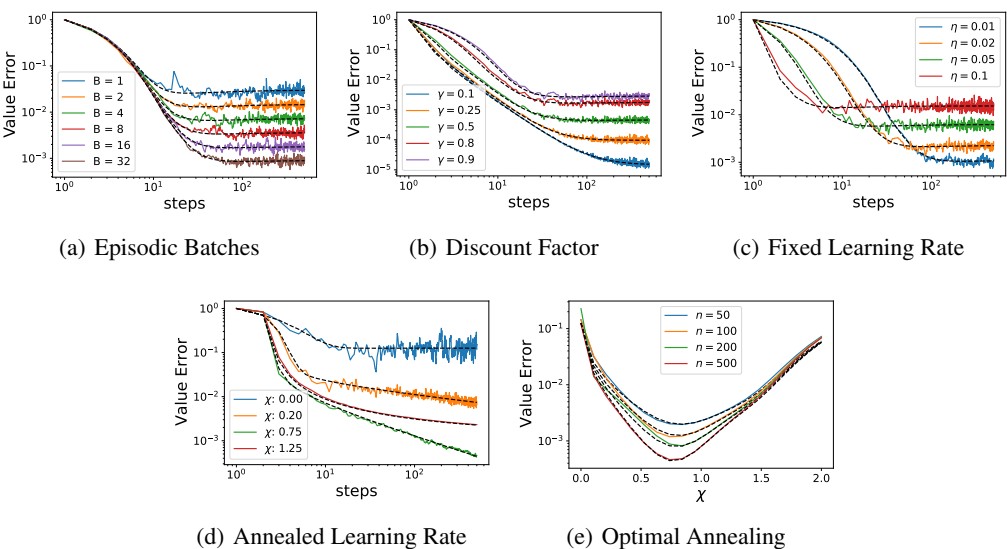

(a) Episodic Batches      (b) Discount Factor      (c) Fixed Learning Rate

(d) Annealed Learning Rate      (e) Optimal Annealing

Figure 3: Finite batch size, discount factor and learning rate all contribute to a stochastic semi-gradient plateau in the TD dynamics. The features are generated from a synthetic power law covariance with exponential temporal autocorrelation (see Appendix G). Dashed black lines are theory. In general, for fixed learning rate $\eta$, the plateau scales as $\mathcal{O}(\eta\gamma^2 B^{-1})$. (a) Larger batch sizes $B$ reduce SGD noise and leads to a lower plateau in the reducible value error for a decoupled power-law feature model. (b) Larger discount factor $\gamma$ and (c) larger learning rate $\eta$ lead to higher SGD plateau floor. (d) An annealing strategy $\eta_n \sim \eta_0 n^{-\chi}$ for $\chi > 0$ can allow one to avoid the plateau. For slow annealing (small $\chi$), the error scales as $\mathcal{L}_n \sim \mathcal{O}(n^{-\chi})$. (e) The value error as a function of the learning rate annealing exponent $\chi$ defined by $\eta_n = \eta_0 n^{-\chi}$. For this task, the optimal exponent balances the scale of the asymptote with the rate of convergence.

The stochastic noise from TD learning has striking qualitative differences from SGD noise in the standard supervised case. In standard supervised learning (such as $\gamma = 0$ version of this theory), the stochastic gradient noise does not prevent the model from fitting the target function with zero error provided the features are sufficiently rich to represent the target function. However, this is not the case in TD learning, where the predicted value $\hat{V}(s)$ is bootstrapped using the model's weights $\boldsymbol{w}_n$ at each iteration $n$. This leads to asymptotic plateaus in learning curves. Our theory can predict these plateaus and their scaling whose proof is given in Appendix B.7.

**Proposition 5.1.** *Our theoretical learning curves exhibit a fixed point for the value error dynamics for finite $B$ and non-zero $\eta$ and $\gamma$. For small $\frac{\eta\gamma^2}{B}$, we deduce that $\boldsymbol{M}$ satisfies a self-consistent*

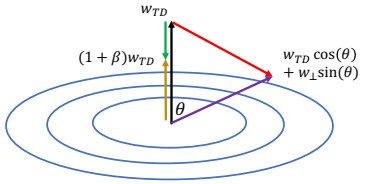

(a) Geometry of Reward Shaping

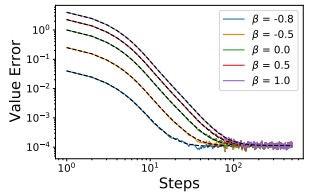

(b) Scale-based Shaping

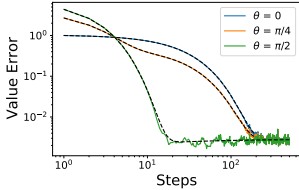

(c) Rotation-based Shaping

Figure 4: The theory can be used to understand how reward shaping decisions alter temporal difference learning dynamics. (a) A visualization of possible reward shaping potentials $\phi(s) = \boldsymbol{w}_\phi \cdot \boldsymbol{\psi}(s)$ strategies in feature space. Probability density level curves for the features are depicted in blue. Reshaping with $\boldsymbol{w}_\phi = \beta \boldsymbol{w}_{TD}$ for scale factor $\beta$ merely changes the scale of weights which must be recovered (gold) and does not change timescales of TD dynamics. (b) The value error dynamics for the scale based reward shaping for the features in Figure 3. On the other hand, rotation based reward shaping where $\boldsymbol{w}_\phi$ is not parallel to $\boldsymbol{w}_V$ (red) leads to a potentially helpful mixture of timescales if the new target vector is more aligned with feature dimensions with high variance (purple). In (c), we plot loss curves for rotation angle $\theta$ between the original mode $\boldsymbol{w}_V$ and the top eigenvector of the feature covariance matrix $\bar{\boldsymbol{\Sigma}}$. Dashed black lines are theory.

asymptotic scaling of the form $\boldsymbol{M} = \mathcal{O}\left(\frac{\eta\gamma^2}{B}\right)$ implying an asymptotic value error scaling of $\mathcal{L} \sim \frac{1}{N} Tr \boldsymbol{M} \bar{\boldsymbol{\Sigma}} \sim \mathcal{O}\left(\frac{\eta\gamma^2}{B}\right)$.

In Figure 3, we demonstrate that our theory predicts the plateaus and their scaling as a function of finite batch size $B$ (Figure 3(a)), non-zero discount factor $\gamma > 0$ (Figure 3(b)) and non-negligible learning rate (Figure 3(c)).

A strategy used in the literature to increase rates of convergence and improve asymptotic behavior is adaptation of the learning learning through an annealing schedule [1, 16, 62, 63]. To overcome this plateau in the loss, we consider annealing the learning rate $\eta_n$ with iteration $n$. In Figure 3(d), we show the effect of annealing the learning rate as a power law $\eta_n = \eta_0 n^{-\chi}$ for some non-negative exponent $\chi$. For $\chi = 0$ the learning rate is constant and a fixed plateau is reached. For small nonzero $\chi$, such as $\chi = 0.2$, the value error is, after an initial transient, always near its instantaneous fixed point plateau so the loss scales linearly with the learning rate, giving the asymptotic rate $\mathcal{L}_n \sim \mathcal{O}(n^{-\chi})$. For large $\chi$, the learning rate decreases very quickly and the plateau is never reached. Our approach can be used to find an optimal annealing exponent $\chi$ and in Figure 3(e), we show that the optimal annealing exponent balances these effects and is well predicted by our theory.

## 6 Reward Shaping

Another strategy to improve the learning dynamics in reinforcement learning algorithms is reward shaping [64]. In standard supervised learning, the goal is to directly approximate the target objective given a cost function. However, in reinforcement learning, the objective is not to estimate rewards at each state directly but the discounted sum of future rewards, the value function. Importantly, many different reward schedules can lead to identical value functions. Reward shaping exploits this symmetry to speed up learning by altering the structure of TD updates and SGD noise. Here, we provide a theoretical description of the changes in the learning dynamics due to reward shaping which suggests they can be understood through a change of the alignment between the original rewards and the reshaped rewards in the space of the features used to represent the states.

The original ideas around reward shaping were inspired by work in experimental psychology and were closer to what is now studied as curriculum learning [65–67]. Reward shaping as currently used in reinforcement learning directly changes the reward function by adding a potential-based shaping function $F$ such that $F(s_t, a, s_{t+1}) = \gamma\phi(s_{t+1}) - \phi(s_t)$ [64]. In each step of the algorithm we feed

the following *reshaped rewards* $\tilde{R}$ to the TD learner

$$\tilde{R}(s_t) = \begin{cases} R(s_t) - \gamma\phi(s_{t+1}) & t = 0 \\ R(s_t) + \phi(s_t) - \gamma\phi(s_{t+1}) & t > 0 \end{cases}. \tag{8}$$

We note that this transformation simply offsets the target value function by $\phi(s)$ as the series above telescopes with a cancellation of $\phi(s_t)$ between the $t-1$ and $t$-th terms [64] (see Appendix C). However, the dynamics of TD learning with these reshaped rewards $\tilde{R}$ is quite distinct from the dynamics with original rewards $R$. Here, we study the case where we can express $\phi(s)$ as a linear function of our features: $\phi(s) = \boldsymbol{\psi}(s) \cdot \boldsymbol{w}_\phi$. This leads to a change in the dynamics for $\boldsymbol{M}_n$ and $\langle \boldsymbol{w}_n \rangle$ that we describe in the Appendix C.

In Figure 4, we illustrate the possible benefits of reward shaping. We explore two types of reward shaping. First, a scale based reward shaping where $\boldsymbol{w}_\phi$ is parallel to the target TD weights $\boldsymbol{w}_{TD}$. This merely changes the overall scale of the weights needed to converge in the dynamics, leading to similar timescales and an identical plateau for TD learning as we show in Figure 4 (b). On the other hand, reward shaping which rotates the fixed point of the TD dynamics into directions of higher feature variance can improve timescales of convergence. In Figure 4 (c), we show an example where we vary the angle $\theta$ of the shaped-TD fixed point (see also Appendix C).

## 7   TD Learning Plateaus in More Realistic Settings

In this section, we test if some of the phenomena observed in our theory and experiments also hold in more realistic settings. We perform TD learning with Fourier features to evaluate a pre-trained policy on MountainCar-v0. As expected, we see that the value error plateaus to an error level determined by both the learning rate (Figure 5a) and batch size (Figure 5b) due to semigradient noise.

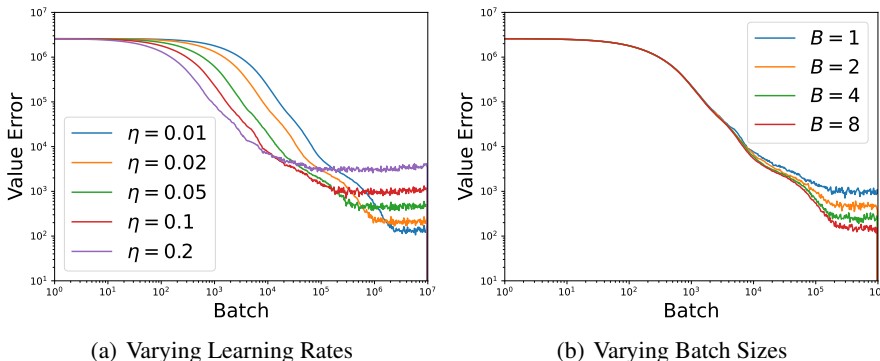

(a) Varying Learning Rates       (b) Varying Batch Sizes

Figure 5: Policy evaluation in MountainCar-v0 environment. The policy was learned with tabular $\epsilon$-greedy Q-learning (see Appendix F for details). (a) Value error curves for different $\eta$ when $B = 1$. (b) Value error curves for different $B$ with $\eta = 0.1$. Shaded area denotes 95% confidence interval over 10 seeds.

We show that the plateaus obey the predicted scalings of $\mathcal{O}(\eta B^{-1})$ in Appendix F.

## 8   Discussion

Our work presents a new approach using concepts from statistical physics to derive average-case learning curve for *policy evaluation* in TD-learning. However, it is only a first step towards a new theory of learning dynamics in reinforcement learning.

One major limitation of the present work is that it concerns linear function approximation where the features representing states/actions are fixed throughout learning. This limit can apply to neural networks in the "lazy" regime of training [68, 69], however it cannot account for neural networks that adapt their internal representations to the structure of the reward function. This differs from the setting of most practical algorithms, including in deep reinforcement learning, that specifically adapt their representations.

Our theory provides a description of learning dynamics through a set of iterative equations (Proposition 3.1). In Figure 1 we evaluate these dynamics for a simple MDP but although the predicted dynamics present an excellent fit to the empirical simulations, the iterative equations can be difficult to interpret and computationally expensive to evaluate in a larger network and more realistic tasks. Nevertheless, our equations can be used to derive some scaling between key parameters of the algorithm for example by studying their fixed points as in Proposition 5.1.

Here, we considered the simplest form of temporal difference learning, batched online TD(0). In future work, it will be important to further characterize the behavior for online TD(0) with batch size $B = 1$ and to expand our approach to TD($\lambda$) and other return distributions. Similarly, expanding our theory to the offline setting, in which the buffer of resampled trajectories would be of finite size, could provide an understanding of how the interactions between parameters govern convergence and divergence [1, 70–72].

Another limitation of our work is that we only considered the setting of *policy evaluation* with a fixed policy. The goal of an RL agent is to learn how to act in the work and not merely to represent the value is its states. Unlike in supervised learning, the changes in the value function affect the policy but in many of RL algorithms, for example in *actor-critic* architecture, there is a separation of the *policy evaluation* (critic) and the *policy learning* (actor) [73, 74]. Such algorithms estimate the value associated with state/action pairs under a given policy and then use this information to make beneficial updates to the policy, usually with the value and policy functions approximated by separate neural networks. In this paper, we only treated the first part of this process. Recently, a related approach has been used to analyse the dynamics of *policy learning* in an "RL perceptron" setup [75]. A full theory of reinforcement learning combining *policy evaluation* and *policy learning* remains difficult due to the interaction between the two processes, but combining these approaches would be fruitful. One promising direction is in settings where the timescales of the two processes are different [76], such as when *policy learning* occurring at a much slower rate which is often the case in practice.

Beyond developing a theory of learning dynamics in reinforcement learning, the approach could be used in neuroscience to understand how neural representation of space or value can shape the learning dynamics at the behavioral level. Ideas from reinforcement learning have been extremely influential to understand phenomena observed in neuroscience and have been mapped directly onto specific brain circuits [77–79]. The place cells of the hippocampus [60] exhibit localized tuning as the example in Figure 1 and together with grid cells in enthorinal cortex are thought to be crucial for navigation in spatial and cognitive spaces and their tuning is shaped by experience [61, 79–81]. Our theory specifically link the structure of representations, policy and reward to learning rates, which can all be experimentally measured simultaneously and could shed on light on how the spectral properties of representations govern learning and navigation [79, 82], similarly to how the mean field theories we have used here can explain learning of sensory features [83]. Future work could straightforwardly extend this DMFT formalism to deal with replay of sampled experiences during TD learning [84] at the cost of tracking correlations of weight updates across iterations of the algorithm [52].

To summarize, our work provide a new promising direction towards a theory of learning dynamics in reinforcement learning in artificial and biological agents.

## Acknowledgments and Disclosure of Funding

BB is supported by a Google PhD Fellowship. CP and BB were supported by NSF grant DMS-2134157. CP is further supported by NSF CAREER Award IIS-2239780, and a Sloan Research Fellowship. PM was supported by NIH grant 5R01DC017311 to Venkatesh Murthy and Naoshige Uchida. HK was supported by the Harvard College Research Program. This work has been made possible in part by a gift from the Chan Zuckerberg Initiative Foundation to establish the Kempner Institute for the Study of Natural and Artificial Intelligence. We thank Jacob Zavatone-Veth for useful discussions and comments on this manuscript.

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

# Appendix

## A  General Convergence Considerations for MDPs in Finite State Space

In this section, we will discuss the infinite batch limit and compare the value function obtained with TD to the ground truth value function. We will, for simplicity, consider in this section a Markov reward process with transition matrix $p(s_{t+1} = s'|s_t = s) = \Pi(s, s')$. The general theory described in the main text does not only apply to MDPs, but the convergence analysis for MDPs is much more straightforward so we describe it here. In this case, the ground truth value function satisfies

$$V(s) = R(s) + \gamma \sum_{s'} \Pi(s, s')V(s') \tag{A.1}$$

which gives the vector equation $\boldsymbol{V} = (\boldsymbol{I} - \gamma\boldsymbol{\Pi})^{-1}\boldsymbol{R}$ for $\boldsymbol{V}, \boldsymbol{R} \in \mathbb{R}^{|\mathcal{S}|}$. Suppose the limiting distribution over states is $\boldsymbol{p} \in \mathbb{R}^{|\mathcal{S}|}$ which has entries $p(s) = \frac{1}{T}\sum_{t=1}^{T} p(s_t = s)$. The fixed point of TD dynamics is

$$\boldsymbol{\Psi}\text{diag}(\boldsymbol{p})\boldsymbol{\Psi}^\top \boldsymbol{w}_{TD} = \boldsymbol{\Psi}\text{diag}(\boldsymbol{p})\boldsymbol{R} + \gamma\boldsymbol{\Psi}\text{diag}(\boldsymbol{p})\boldsymbol{\Pi}\boldsymbol{\Psi}^\top \boldsymbol{w}_{TD}. \tag{A.2}$$

We now consider the two possible cases for this fixed point condition.

**Case 1: Underparameterized Regime**  First, if the feature dimension $N$ is smaller than the size of the state space $|\mathcal{S}|$ and the features are maximal rank, then the TD learning fixed point is

$$\boldsymbol{w}_{TD} = \left(\boldsymbol{\Psi}\text{diag}(\boldsymbol{p})\boldsymbol{\Psi}^\top - \gamma\boldsymbol{\Psi}\text{diag}(\boldsymbol{p})\boldsymbol{\Pi}\boldsymbol{\Psi}^\top\right)^{-1}\boldsymbol{\Psi}\text{diag}(\boldsymbol{p})\boldsymbol{R} \tag{A.3}$$

In this case, the value function is not learned perfectly, as can be seen by computing $\hat{\boldsymbol{V}} = \boldsymbol{\Psi}^\top \boldsymbol{w}_{TD}$ and comparing to the ground truth $\boldsymbol{V} = (\boldsymbol{I} - \gamma\boldsymbol{\Pi})^{-1}\boldsymbol{R}$. In this case, we would say that TD learning has an *irreducible value error* due to capturing only a $N$ dimensional projection of the value function.

**Case 2: Overparameterized Regime**  Alternatively, if the feature dimension exceeds the total number of states, then the fixed point equation for TD is underspecified. However, throughout TD learning $\boldsymbol{w}_{TD} \in \text{span}\{\boldsymbol{\psi}(s)\}_{s \in \mathcal{S}}$ so we can instead consider the decompostion $\boldsymbol{w}_V = \sum_s \alpha(s)\boldsymbol{\psi}(s)$, where $\boldsymbol{\alpha} \in \mathbb{R}^{|\mathcal{S}|}$ satisfies

$$\text{diag}(\boldsymbol{p})(\boldsymbol{I} - \gamma\boldsymbol{\Pi})\boldsymbol{K}\boldsymbol{\alpha} = \text{diag}(\boldsymbol{p})\boldsymbol{R} \tag{A.4}$$

where $\boldsymbol{K} \in \mathbb{R}^{|\mathcal{S}| \times |\mathcal{S}|}$ is the kernel computed with features $K(s, s') = \boldsymbol{\psi}(s) \cdot \boldsymbol{\psi}(s')$. The solution to the above equation is unique and the learned value function $\hat{\boldsymbol{V}} = \boldsymbol{\Psi}^\top \boldsymbol{w}_{TD} = \boldsymbol{K}\boldsymbol{K}^{-1}(\boldsymbol{I} - \gamma\boldsymbol{\Pi})^{-1}\boldsymbol{R} = (\boldsymbol{I} - \gamma\boldsymbol{\Pi})^{-1}\boldsymbol{R} = \boldsymbol{V}$. Therefore, in the over-parameterized limit, the irreducible value error for TD learning is zero. This limit was considered dynamically in the infinite batch (vanishing SGD noise) setting by [44].

## B  Derivation of Learning Curves

In this section, we now consider the dynamics of TD learning when $B$ random episodes are sampled at a time. In this calculation, the finite batch of episodes leads to non-negligible SGD effects which can cause undesirable plateaus in TD dynamics.

### B.1  Field Theory Derivation

In this section we use a Gaussian field theory formalism to compute the learning curve in the high dimensional asymptotic limit $N, B \to \infty$ with $B/N = \alpha$. The episode length $T$ is treated as $\mathcal{O}(1)$. While this paper focuses on the online setting, where fresh trajectories $\{\tau_n^\mu\}$ are sampled at each iteration $n$, this model can be straightforwardly extended to the case where a fixed number of experience trajectories $\{\tau^\mu\}$ are replayed repeatedly during TD learning. We leave the experience

replay dynamic mean field theory calculation for future work. The starting point of our analysis is tracking the moment generating function for the iterate dynamics

$$Z[\{\boldsymbol{j}_n\}] = \mathbb{E}_{\{\boldsymbol{w}_n\},\{s_n^\mu(t)\}} \exp\left(i\sum_{n=0}^\infty \boldsymbol{j}_n \cdot \boldsymbol{w}_n\right). \tag{B.1}$$

To compute this object over random draws of training trajectories, we express the joint average over $\boldsymbol{w}_n, \{s_n^\mu(t)\}$ into conditional averages over $\boldsymbol{w}_n, \{\Delta_n^\mu(t)\}|\{\boldsymbol{\psi}_n^\mu(t)\}$. To simplify the computation, in this section, we will compute the learning curve for mean zero features $\boldsymbol{\mu}(s) = 0$ and

$$Z = \mathbb{E}_{\{\boldsymbol{\psi}_n^\mu(t)\}} \int \prod_n d\boldsymbol{w}_n \delta\left(\boldsymbol{w}_{n+1} - \boldsymbol{w}_n - \frac{\eta}{\sqrt{BT}} \sum_{\mu t} \Delta_n^\mu(t)\boldsymbol{\psi}_n^\mu(t)\right) \exp\left(i\sum_{n=0}^\infty \boldsymbol{j}_n \cdot \boldsymbol{w}_n\right)$$

$$\times \int \prod_{t\mu n} d\Delta_n^\mu(t) \, \delta\left(\Delta_n^\mu(t) - \frac{1}{\sqrt{N}}(\boldsymbol{w}_R - \boldsymbol{w}_n) \cdot \boldsymbol{\psi}_n^\mu(t) - \frac{\gamma}{\sqrt{N}}\boldsymbol{w}_n \cdot \boldsymbol{\psi}_n^\mu(t+1)\right) \tag{B.2}$$

Expressing the Dirac-delta function as a Fourier integral $\delta(z) = \int \frac{d\hat{z}}{2\pi} \exp(i\hat{z}z)$ for each of our constraints. Under the *Gaussian equivalence ansatz*, we can easily average over Gaussian $\psi$ to obtain

$$Z = \int \mathcal{D}\Delta \mathcal{D}\hat{\Delta}\mathcal{D}\boldsymbol{w}\mathcal{D}\hat{\boldsymbol{w}} \exp\left(-\frac{\eta^2}{2BT^2} \sum_{n\mu}\sum_{tt'} \Delta_n^\mu(t)\Delta_n^\mu(t')\hat{\boldsymbol{w}}_n^\top \boldsymbol{\Sigma}(t,t')\hat{\boldsymbol{w}}_n\right)$$

$$\exp\left(i\sum_n \hat{\boldsymbol{w}}_n \cdot (\boldsymbol{w}_{n+1} - \boldsymbol{w}_n)\right)$$

$$\exp\left(-\frac{1}{2N} \sum_{n\mu tt'} \left[(\boldsymbol{w}_R - \boldsymbol{w}_n)\hat{\Delta}_n^\mu(t)\right] \boldsymbol{\Sigma}(t,t') \left[(\boldsymbol{w}_R - \boldsymbol{w}_n)\hat{\Delta}_n^\mu(t')\right]\right)$$

$$\exp\left(-\frac{\gamma^2}{2N} \sum_{n\mu tt'} \hat{\Delta}_n^\mu(t-1)\hat{\Delta}_n^\mu(t'-1)\boldsymbol{w}_n^\top \boldsymbol{\Sigma}(t,t')\boldsymbol{w}_n\right)$$

$$\exp\left(-\frac{\gamma}{N} \sum_{n\mu tt'} \hat{\Delta}_n^\mu(t-1)\hat{\Delta}_n^\mu(t')\boldsymbol{w}_n^\top \boldsymbol{\Sigma}(t,t')(\boldsymbol{w}_R - \boldsymbol{w}_n)\right)$$

$$\exp\left(-\frac{\eta}{\sqrt{NBT}} \sum_{n\mu tt'} \left[\hat{\Delta}_n^\mu(t)(\boldsymbol{w}_R - \boldsymbol{w}_n) + \gamma\hat{\Delta}_n^\mu(t-1)\boldsymbol{w}_n\right]^\top \boldsymbol{\Sigma}(t,t')\hat{\boldsymbol{w}}_n\Delta_n^\mu(t')\right)$$

$$\exp\left(i\sum_{n\mu t} \hat{\Delta}_n^\mu(t)\Delta_n^\mu(t) + i\sum_n \boldsymbol{j}_n \cdot \boldsymbol{w}_n\right) \tag{B.3}$$

where we adopted the shorthand $\mathcal{D}\Delta = \prod_{\mu,n,t} d\Delta_n^\mu(t)$ for the measure for the collection of variables $\{\Delta_n^\mu(t)\}$. Likewise one should interpret $\mathcal{D}\boldsymbol{w} = \prod_n d\boldsymbol{w}_n$. To analyze the high dimensional limit of the above moment generating function, we introduce order parameters for the theory

$$Q_n(t,t') = \frac{1}{B}\sum_{\mu=1}^B \Delta_n^\mu(t)\Delta_n^\mu(t') \,,\; C_n(t,t') = \frac{1}{N}\boldsymbol{w}_n^\top \boldsymbol{\Sigma}(t,t')\boldsymbol{w}_n$$

$$C_n^R(t,t') = \frac{1}{N}\boldsymbol{w}_R\boldsymbol{\Sigma}(t,t')\boldsymbol{w}_n \,,\; D_n(t,t') = -\frac{i}{N}\hat{\boldsymbol{w}}_n^\top \boldsymbol{\Sigma}(t,t')\boldsymbol{w}_n \,,\; D_n^R(t,t') = -\frac{i}{N}\hat{\boldsymbol{w}}_n^\top \boldsymbol{\Sigma}(t,t')\boldsymbol{w}_R \tag{B.4}$$

For each of these order parameters, we enforce the definition of the order parameter using the Fourier representation of a Dirac-delta function

$$
1 = B \int dQ_n(t,t')\delta\left( BQ_n(t,t') - \sum_\mu \Delta_n^\mu(t)\Delta_n^\mu(t') \right)
$$

$$
= B \int \frac{dQ_n(t,t')d\hat{Q}_n(t,t')}{4\pi i} \exp\left( \frac{B}{2}\hat{Q}_n(t,t')Q_n(t,t') - \frac{1}{2}\sum_\mu \Delta_n^\mu(t)\Delta_n^\mu(t')\hat{Q}_n(t,t') \right).
$$
(B.5)

Repeating this procedure for all order parameters $q = \{Q, \hat{Q}, C, \hat{C}, C^R, \hat{C}^R, D, \hat{D}, D^R, \hat{D}^R\}$ and disregarding irrelevant prefactors, we have the following formula for the moment generating function

$$
Z \propto \int \mathcal{D}q \exp\left( \frac{N}{2}S[q] \right)
$$
(B.6)

where the action $S$ has the form

$$
S = \sum_n \sum_{tt'} \left[ \alpha Q_n(t,t')\hat{Q}_n(t,t') + C_n(t,t')\hat{C}_n(t,t') + C_n^R(t,t')\hat{C}_n^R(t,t') \right]
$$

$$
-2\sum_n \sum_{tt'} \left[ D_n(t,t')\hat{D}_n(t,t') + D_n^R(t,t')\hat{D}_n^R(t,t') \right] + \frac{2}{N}\ln \mathcal{Z}_w + 2\alpha \ln \mathcal{Z}_\Delta
$$

$$
\mathcal{Z}_w = \int \mathcal{D}\boldsymbol{w}\mathcal{D}\hat{\boldsymbol{w}} \exp\left( -\frac{\eta^2}{2T^2}\sum_{ntt'} Q_n(t,t')\hat{\boldsymbol{w}}_n^\top \boldsymbol{\Sigma}(t,t')\hat{\boldsymbol{w}}_n + i\sum_n \hat{\boldsymbol{w}}_n \cdot (\boldsymbol{w}_{n+1} - \boldsymbol{w}_n) \right)
$$

$$
\exp\left( -\frac{1}{2}\sum_{ntt'} \hat{C}_n(t,t')\boldsymbol{w}_n^\top \boldsymbol{\Sigma}(t,t')\boldsymbol{w}_n - \frac{1}{2}\hat{C}_n^R(t,t')\boldsymbol{w}_R^\top \boldsymbol{\Sigma}(t,t')\boldsymbol{w}_n \right)
$$

$$
\exp\left( -i\sum_{ntt'} \hat{D}_n(t,t')\hat{\boldsymbol{w}}_n^\top \boldsymbol{\Sigma}(t,t')\boldsymbol{w}_n - i\sum_{ntt'} \hat{D}_n^R(t,t')\hat{\boldsymbol{w}}_n^\top \boldsymbol{\Sigma}(t,t')\boldsymbol{w}_R \right)
$$

$$
\mathcal{Z}_\Delta = \int \mathcal{D}\Delta\mathcal{D}\hat{\Delta} \exp\left( -\frac{1}{2}\sum_{ntt'} \hat{Q}_n(t,t')\Delta_n(t)\Delta_n(t') + i\sum_{nt} \hat{\Delta}_n(t)\Delta_n(t) \right)
$$

$$
\exp\left( -\frac{1}{2}\sum_{ntt'} \hat{\Delta}_n(t)\hat{\Delta}_n(t')\left[ \frac{1}{N}\boldsymbol{w}_R^\top \boldsymbol{\Sigma}(t,t')\boldsymbol{w}_R + C(t,t') \right] \right)
$$

$$
\exp\left( \frac{1}{2}\sum_{ntt'} \hat{\Delta}_n(t)\hat{\Delta}_n(t')\left[ C^R(t,t') + C^R(t',t) \right] \right)
$$

$$
\exp\left( -\gamma \sum_{t,t'} \hat{\Delta}_n(t)\hat{\Delta}_n(t'-1)C_n^R(t,t') \right)
$$

$$
\exp\left( -\frac{\gamma^2}{2}\sum_{t,t'} \hat{\Delta}_n(t-1)\hat{\Delta}_n(t'-1)C_n(t,t') \right)
$$

$$
\exp\left( -\frac{\eta i}{\sqrt{\alpha}T}\sum_{nt,t'} \hat{\Delta}_n(t)\left[ D_n^R(t',t) - D_n(t',t) + \gamma D_n(t',t+1) \right]\Delta_n(t') \right)
$$
(B.7)

The function $\mathcal{Z}$ has the interpretation of an effective partition function conditional on order parameters $q$. To study the $N \to \infty$ limit, we use the steepest descent method and analyze the saddle point

$\frac{\partial S}{\partial q} = 0$. These saddle point equations give

$$\frac{\partial S}{\partial \hat{Q}_n(t,t')} = \alpha Q_n(t,t') - \alpha \left\langle \Delta_n(t) \Delta_n(t') \right\rangle = 0$$

$$\frac{\partial S}{\partial Q_n(t,t')} = \alpha \hat{Q}_n(t,t') - \frac{\eta^2}{T^2 N} \left\langle \hat{\boldsymbol{w}}_n^\top \boldsymbol{\Sigma}(t,t') \hat{\boldsymbol{w}}_n \right\rangle = 0$$

$$\frac{\partial S}{\partial \hat{C}_n(t,t')} = C_n(t,t') - \frac{1}{N} \left\langle \boldsymbol{w}_n^\top \boldsymbol{\Sigma}(t,t') \boldsymbol{w}_n \right\rangle = 0$$

$$\frac{\partial S}{\partial C_n(t,t')} = \hat{C}_n(t,t') - \alpha \left\langle \hat{\Delta}_n(t) \hat{\Delta}_n(t') + \gamma^2 \hat{\Delta}_n(t-1) \hat{\Delta}_n(t'-1) \right\rangle = 0$$

$$\frac{\partial S}{\partial \hat{C}_n^R(t,t')} = C_n^R(t,t') - \frac{1}{N} \left\langle \boldsymbol{w}_R^\top \boldsymbol{\Sigma}(t,t') \boldsymbol{w}_n \right\rangle = 0$$

$$\frac{\partial S}{\partial C_n(t,t')} = \hat{C}_n(t,t') - \alpha \left\langle \hat{\Delta}_n(t) \hat{\Delta}_n(t') + \gamma \hat{\Delta}_n(t) \hat{\Delta}_n(t'-1) \right\rangle = 0$$

$$\frac{\partial S}{\partial \hat{D}_n(t,t')} = -2D_n(t,t') - \frac{2i}{N} \left\langle \hat{\boldsymbol{w}}_n^\top \boldsymbol{\Sigma}(t,t') \boldsymbol{w}_n \right\rangle = 0$$

$$\frac{\partial S}{\partial \hat{D}_n^R(t,t')} = -2D_n^R(t,t') - \frac{2i}{N} \left\langle \hat{\boldsymbol{w}}_n^\top \boldsymbol{\Sigma}(t,t') \boldsymbol{w}_n \right\rangle = 0$$

$$\frac{\partial S}{\partial D_n(t,t')} = -2\hat{D}_n(t,t') - \frac{2\alpha\eta i}{\sqrt{\alpha}T} \left\langle \gamma \hat{\Delta}_n(t-1) \Delta_n(t') - \hat{\Delta}_n(t) \Delta_n(t') \right\rangle = 0$$

$$\frac{\partial S}{\partial D_n^R(t,t')} = -2\hat{D}_n^R(t,t') - \frac{2\alpha\eta i}{\sqrt{\alpha}T} \left\langle \hat{\Delta}_n(t) \Delta_n(t') \right\rangle = 0 \tag{B.8}$$

The brackets $\langle \rangle$ denote averaging over the stochastic processes defined by moment generating functions $\mathcal{Z}_\Delta, \mathcal{Z}_w$. After these saddle point equations are solved the order parameters $q$ are treated as non-random and a Hubbard-Stratonovich transformation is employed. For example,

$$\exp\left(-\frac{1}{2}\hat{\boldsymbol{w}}_n \left[ \frac{\eta^2}{T^2} \sum_{tt'} Q_n(t,t') \boldsymbol{\Sigma}(t,t') \right] \hat{\boldsymbol{w}}_n \right) = \mathbb{E}_{\boldsymbol{u}_n^w} \exp\left( i \sum_n \boldsymbol{u}_n^w \cdot \hat{\boldsymbol{w}}_n \right) \tag{B.9}$$

where the average is over $\boldsymbol{u}_n^w \sim \mathcal{N}\left(0, \eta^2 T^{-2} \sum_{tt'} Q_n(t,t') \boldsymbol{\Sigma}(t,t')\right)$. After introducing these Hubbard fields $\boldsymbol{u}_n^w$ and $u_n^\Delta(t)$, we can perform the integrals over $\hat{\boldsymbol{w}}_n$ and $\hat{\Delta}_n(t)$ which collapse to Dirac-Delta functions. The resulting identities of the delta functions define the following stochastic processes on $\boldsymbol{w}_n$ and $u_n^\Delta$

$$\boldsymbol{w}_{n+1} = \boldsymbol{w}_n + \boldsymbol{u}_n^w + \sum_{tt'} \hat{D}_n^R(t,t') \boldsymbol{\Sigma}(t,t') \boldsymbol{w}_R + \sum_{t,t'} \hat{D}_n(t,t') \boldsymbol{\Sigma}(t,t') \boldsymbol{w}_n$$

$$\Delta_n(t) = u_n^\Delta(t) + \frac{\eta}{\sqrt{\alpha}T} \sum_{tt'} [D_n^R(t,t') - D_n(t,t') - \gamma D_n(t',t+1)] \Delta_n(t'). \tag{B.10}$$

Using a similar trick, we can show that for any observable depending on $\boldsymbol{w}_n$ or $\{\Delta_n(t)\}$ that

$$-i \left\langle \hat{\boldsymbol{w}}_n O(\boldsymbol{w}_n) \right\rangle = \left\langle \frac{\partial}{\partial \boldsymbol{u}_n} O(\boldsymbol{w}_n) \right\rangle$$

$$-i \left\langle \hat{\Delta}_n(t) O(\{\Delta_n(t')\}) \right\rangle = \left\langle \frac{\partial}{\partial u_n^\Delta(t)} O(\{\Delta_n(t')\}) \right\rangle \tag{B.11}$$

Since $\boldsymbol{w}_n$ is independent. This can be used to conclude

$$D_n(t,t') = 0 \,, \quad D_n^R(t,t') = 0 \tag{B.12}$$

which implies that $\Delta_n(t) = u_n^\Delta(t)$. Consequently the response functions have trivial structure

$$\hat{D}_n(t) = -\frac{\eta\sqrt{\alpha}}{T} \left[ \delta(t-t') - \gamma\delta(t-1-t') \right] \,, \quad \hat{D}_n^R(t,t') = \frac{\sqrt{\alpha}\eta}{T} \delta(t-t'). \tag{B.13}$$

We therefore obtain a stochastic process of the form

$$\boldsymbol{w}_{n+1} = \boldsymbol{w}_n + \boldsymbol{u}_n^w + \frac{\eta\sqrt{\alpha}}{T} \sum_t \boldsymbol{\Sigma}(t,t)\boldsymbol{w}_R - \frac{\eta\sqrt{\alpha}}{T} \sum_t \left[\boldsymbol{\Sigma}(t,t) - \gamma\boldsymbol{\Sigma}(t,t+1)\right]\boldsymbol{w}_n$$

$$\boldsymbol{u}_n \sim \mathcal{N}\left(0, \frac{\eta^2}{T^2}\sum_{tt'} Q_n(t,t')\boldsymbol{\Sigma}(t,t')\right) \ , \ \{\Delta_n(t)\} \sim \mathcal{N}(0, \boldsymbol{Q}_n)$$

$$Q_n(t,t') = \langle \Delta_n(t)\Delta_n(t')\rangle = \frac{1}{N}\boldsymbol{w}_R\boldsymbol{\Sigma}(t,t')\boldsymbol{w}_R - C^R(t,t') - C^R(t',t) + C(t,t')$$

$$C_n(t,t') = \frac{1}{N}\left\langle \boldsymbol{w}_n^\top \boldsymbol{\Sigma}(t,t')\boldsymbol{w}_n\right\rangle \ , \ C_n^R(t,t') = \frac{1}{N}\left\langle \boldsymbol{w}_R^\top \boldsymbol{\Sigma}(t,t')\boldsymbol{w}_n\right\rangle$$

These are the final equations defining the stochastic evolution of $\boldsymbol{w}_n$ and $\Delta_n(t)$.

## B.2   Simplifying the Saddle Point Equations

Using the above saddle point equations, we see that the variables $\{\Delta_n(t)\}$ and $\{\boldsymbol{w}_n\}$ will be Gaussian random variables. It thus suffices to track their mean and covariance. The $\{\Delta_n(t)\}$ variables have zero mean and covariance given by the $Q_n(t,t')$ function. The $\{\boldsymbol{w}_n\}$ variables have the following mean evolution

$$\begin{aligned}
\langle\boldsymbol{w}_{n+1}\rangle &= \langle\boldsymbol{w}_n\rangle + \eta\sqrt{\alpha}\left[\bar{\boldsymbol{\Sigma}}\boldsymbol{w}_R - \left[\bar{\boldsymbol{\Sigma}} - \gamma\bar{\boldsymbol{\Sigma}}_+\right]\langle\boldsymbol{w}_n\rangle\right] \\
&= \langle\boldsymbol{w}_n\rangle + \eta\sqrt{\alpha}\left[\bar{\boldsymbol{\Sigma}} - \gamma\bar{\boldsymbol{\Sigma}}_+\right]\left[\boldsymbol{w}_{TD} - \langle\boldsymbol{w}_n\rangle\right]
\end{aligned} \tag{B.14}$$

where $\boldsymbol{w}_{TD} = \left[\bar{\boldsymbol{\Sigma}} - \gamma\bar{\boldsymbol{\Sigma}}_+\right]^{-1}\bar{\boldsymbol{\Sigma}}\boldsymbol{w}_R$ is the fixed point of the TD dynamics. We next compute $\boldsymbol{M}_n = \left\langle (\boldsymbol{w}_n - \boldsymbol{w}_{TD})(\boldsymbol{w}_n - \boldsymbol{w}_{TD})^\top\right\rangle$ which admits the recursion

$$\boldsymbol{M}_{n+1} = \left(\boldsymbol{I} - \eta\sqrt{\alpha}\left[\bar{\boldsymbol{\Sigma}} - \gamma\bar{\boldsymbol{\Sigma}}_+\right]\right)\boldsymbol{M}_n\left(\boldsymbol{I} - \eta\sqrt{\alpha}\left[\bar{\boldsymbol{\Sigma}} - \gamma\bar{\boldsymbol{\Sigma}}_+\right]\right) + \frac{\eta^2}{T^2}\sum_{tt'}Q_n(t,t')\boldsymbol{\Sigma}(t,t') \tag{B.15}$$

To obtain our formulas which hold for finite batch size, we rescale the learning rate by $\eta \to \eta/\sqrt{\alpha}$ giving the following evolution

$$\langle\boldsymbol{w}_{n+1}\rangle = \langle\boldsymbol{w}_n\rangle + \eta\left[\bar{\boldsymbol{\Sigma}} - \gamma\bar{\boldsymbol{\Sigma}}_+\right]\left[\boldsymbol{w}_{TD} - \langle\boldsymbol{w}_n\rangle\right]$$

$$\boldsymbol{M}_{n+1} = \left(\boldsymbol{I} - \eta\left[\bar{\boldsymbol{\Sigma}} - \gamma\bar{\boldsymbol{\Sigma}}_+\right]\right)\boldsymbol{M}_n\left(\boldsymbol{I} - \eta\left[\bar{\boldsymbol{\Sigma}} - \gamma\bar{\boldsymbol{\Sigma}}_+\right]\right)^\top + \frac{\eta^2}{T^2\alpha^2}\sum_{tt'}Q_n(t,t')\boldsymbol{\Sigma}(t,t') \tag{B.16}$$

After this rescaling, we see that the mean evolution for $\boldsymbol{w}_n$ is independent of $\alpha$ but that the variance picks up an additive term on each step on the order of $\mathcal{O}(\eta^2\alpha^{-2})$ which vanishes in the infinite batch limit $B/N \to \infty$. The error for value learning can be obtained from $\boldsymbol{M}_n$ with $\mathcal{L}_n = \frac{1}{N}\text{Tr}\boldsymbol{M}_n\bar{\boldsymbol{\Sigma}}$. Lastly, we note that we can express the formula for $Q_n(t,t')$ entirely in terms of $\boldsymbol{M}_n$ and $\langle\boldsymbol{w}_n\rangle$. This

gives the lengthy expression

$$
\begin{aligned}
Q_n(t,t') &= \frac{1}{N}\left\langle (\boldsymbol{w}_R - \boldsymbol{w}_n)^\top \boldsymbol{\Sigma}(t,t')(\boldsymbol{w}_R - \boldsymbol{w}_n)\right\rangle + \frac{\gamma}{N}\left\langle (\boldsymbol{w}_R - \boldsymbol{w}_n)^\top \boldsymbol{\Sigma}(t,t'+1)\boldsymbol{w}_n\right\rangle \\
&\quad + \frac{\gamma}{N}\left\langle \boldsymbol{w}_n^\top \boldsymbol{\Sigma}(t+1,t')(\boldsymbol{w}_R - \boldsymbol{w}_n)\right\rangle + \frac{\gamma^2}{N}\left\langle \boldsymbol{w}_n^\top \boldsymbol{\Sigma}(t+1,t'+1)\boldsymbol{w}_n\right\rangle \\
&= \frac{1}{N}\mathrm{Tr}\boldsymbol{M}_n\boldsymbol{\Sigma}(t,t') + \frac{1}{N}\left(\boldsymbol{w}_{TD} - \langle\boldsymbol{w}_n\rangle\right)\left[\boldsymbol{\Sigma}(t,t') + \boldsymbol{\Sigma}(t',t)\right]\left(\boldsymbol{w}_R - \boldsymbol{w}_{TD}\right) \\
&\quad + \frac{1}{N}\left(\boldsymbol{w}_R - \boldsymbol{w}_{TD}\right)^\top \boldsymbol{\Sigma}(t,t')\left(\boldsymbol{w}_R - \boldsymbol{w}_{TD}\right) \\
&\quad - \frac{\gamma}{N}\mathrm{Tr}\boldsymbol{M}_n\left[\boldsymbol{\Sigma}(t,t'+1) + \boldsymbol{\Sigma}(t+1,t')\right] \\
&\quad + \frac{\gamma}{N}\left(\boldsymbol{w}_{TD} - \langle\boldsymbol{w}_n\rangle\right)\left[\boldsymbol{\Sigma}(t,t'+1) + \boldsymbol{\Sigma}(t+1,t')\right]\boldsymbol{w}_{TD} \\
&\quad + \frac{\gamma}{N}\left(\boldsymbol{w}_R - \boldsymbol{w}_{TD}\right)^\top\left[\boldsymbol{\Sigma}(t,t'+1) + \boldsymbol{\Sigma}(t+1,t')\right]\langle\boldsymbol{w}_n\rangle \\
&\quad + \frac{\gamma^2}{N}\mathrm{Tr}\boldsymbol{M}_n\boldsymbol{\Sigma}(t+1,t'+1) + \frac{2\gamma^2}{N}\left(\langle\boldsymbol{w}_n\rangle - \boldsymbol{w}_{TD}\right)\boldsymbol{\Sigma}(t+1,t'+1)\boldsymbol{w}_{TD} \\
&\quad + \frac{\gamma^2}{N}\boldsymbol{w}_{TD}^\top\boldsymbol{\Sigma}(t+1,t'+1)\boldsymbol{w}_{TD}
\end{aligned}
\tag{B.17}
$$

## B.3 Final Result

Below we state in compact form the full final result for our TD learning curves. The below equations give the evolution of the first and second moments of $\boldsymbol{w}_n$ obtained from the mean-field density of the previous section. Concretely, these moments obey dynamics

$$
\langle\boldsymbol{w}_{n+1}\rangle = \langle\boldsymbol{w}_n\rangle + \eta\left[\bar{\boldsymbol{\Sigma}} - \gamma\bar{\boldsymbol{\Sigma}}_+\right]\left[\boldsymbol{w}_V - \langle\boldsymbol{w}_n\rangle\right]
$$

$$
\boldsymbol{M}_{n+1} = \left[\boldsymbol{I} - \eta\bar{\boldsymbol{\Sigma}} + \eta\gamma\bar{\boldsymbol{\Sigma}}_+\right]\boldsymbol{M}_n\left[\boldsymbol{I} - \eta\bar{\boldsymbol{\Sigma}} + \eta\gamma\bar{\boldsymbol{\Sigma}}_+\right]^\top + \frac{\eta^2}{\alpha^2 T^2}\sum_{tt'}Q_n(t,t')\boldsymbol{\Sigma}(t,t')
$$

$$
\begin{aligned}
Q_n(t,t') &= \frac{1}{N}\left\langle (\boldsymbol{w}_R - \boldsymbol{w}_n)^\top \boldsymbol{\Sigma}(t,t')(\boldsymbol{w}_R - \boldsymbol{w}_n)\right\rangle + \frac{\gamma}{N}\left\langle (\boldsymbol{w}_R - \boldsymbol{w}_n)^\top \boldsymbol{\Sigma}(t,t'+1)\boldsymbol{w}_n\right\rangle \\
&\quad + \frac{\gamma}{N}\left\langle \boldsymbol{w}_n^\top \boldsymbol{\Sigma}(t+1,t')(\boldsymbol{w}_R - \boldsymbol{w}_n)\right\rangle + \frac{\gamma^2}{N}\left\langle \boldsymbol{w}_n^\top \boldsymbol{\Sigma}(t+1,t'+1)\boldsymbol{w}_n\right\rangle.
\end{aligned}
\tag{B.18}
$$

These equations can be solved iteratively for $\bar{\boldsymbol{w}}_n, \boldsymbol{M}_n, Q_n$. Finite dimensional versions of this result can be obtained by replacing $\alpha$ with $B/N$ as written in the main text. The value estimation error is

$$
\mathcal{L}_n = \frac{1}{N}\mathrm{Tr}\boldsymbol{M}_n\bar{\boldsymbol{\Sigma}}.
\tag{B.19}
$$

## B.4 Non-Zero Mean Feature

We can also simply modify the DMFT equations if the mean feature is nonvanishing $\boldsymbol{\mu}(s) \neq 0$. In this case, when averaging over all possible trajectories through state space, there is a mean feature vector at each episodic time $\boldsymbol{\mu}(t)$. The above equations are exact for non-zero mean features if $\boldsymbol{\Sigma}(t,t')$ is regarded as the (non-centered) correlation matrix $\langle\boldsymbol{\psi}(t)\boldsymbol{\psi}(t')\rangle$.

## B.5 Action Dependent Rewards

### B.5.1 Expected Q-Learning Reduces to Previous Model

In the case where we consider using features that depend on both states and actions $\boldsymbol{\psi}(s,a)$ then we can use expected value learning to identify the expected value of a state-action pair under policy $\pi$

$$
V(s,a) = R(s,a) + \gamma\left\langle V(s',a')\right\rangle_{s',a'|s,a}
\tag{B.20}
$$

This $V$ function quantifies the expected reward associated with taking action $a$ when in state $s$ and subsequently following policy $\pi$. This problem is structurally identical to the state dependent case

by recognizing that state action pairs $(s, a)$ act as new states $\tilde{s}$. As before, the policy defines the probability distribution over transitions on $\tilde{s}$. We can thus use Equation (4) to calculate the learning curve for this problem.

### B.5.2 Action Dependence Generates Target Noise in State Dependent Value Learning

In the case where the rewards depend on both state and action $R(s, a)$ but features only depend on state $\psi(s)$, we need a slight modification of our theory which models the reward at each state as a mean value (over actions) plus a noise. For each state, we decompose the reward function into mean and fluctuation

$$R(s, a) = \bar{R}(s) + \epsilon(s, a) \, , \; \bar{R}(s) = \mathbb{E}_{a \sim \pi(a|s)} R(s, a) \tag{B.21}$$

The function $\bar{R}(s)$ can again be decomposed into the basis of features $\psi(s)$. However, we need to consider the correlation structure of $\epsilon(s, a)$.

$$\mathbb{E}_\tau \epsilon(s_t, a_t) \epsilon(s_{t'}, a_{t'}) = \mathbb{E}_{s_t} \mathbb{E}_{a_t|s_t} \epsilon(s_t, a_t) \left[ \mathbb{E}_{s_{t'}|s_t, a_t} \left( \mathbb{E}_{a_{t'}|s_{t'}} \epsilon(s_{t'}, a_{t'}) \right) \right]$$

$$= \delta_{t,t'} \mathrm{Var}_{a|s_t} R(s_t, a). \tag{B.22}$$

The above average vanishes for $t \neq t'$ since $\epsilon(s_{t'}, a_{t'})$ is zero mean over $a|s$. We introduce the notation $\sigma_t^2 = \mathrm{Var}_{a|s_t} R(s_t, a)$. Thus, we effectively have a model where our TD errors obey

$$\Delta(t) = \bar{R}(s_t) + \epsilon(s_t, a_t) + \gamma \hat{V}(s_t) - \hat{V}(s_t) \tag{B.23}$$

The addition of this term leads to a simple modification of our $Q(t, t)$ function

$$Q_n(t, t') = \frac{1}{N} \left\langle (\boldsymbol{w}_R - \boldsymbol{w}_n)^\top \boldsymbol{\Sigma}(t, t')(\boldsymbol{w}_R - \boldsymbol{w}_n) \right\rangle + \frac{\gamma}{N} \left\langle (\boldsymbol{w}_R - \boldsymbol{w}_n)^\top \boldsymbol{\Sigma}(t, t' + 1)\boldsymbol{w}_n \right\rangle$$

$$+ \frac{\gamma}{N} \left\langle \boldsymbol{w}_n^\top \boldsymbol{\Sigma}(t + 1, t')(\boldsymbol{w}_R - \boldsymbol{w}_n) \right\rangle + \frac{\gamma^2}{N} \left\langle \boldsymbol{w}_n^\top \boldsymbol{\Sigma}(t + 1, t' + 1)\boldsymbol{w}_n \right\rangle$$

$$+ \delta_{t,t'} \sigma_t^2. \tag{B.24}$$

This change to the $Q_n(t, t')$ correlation function alters the dynamics of $\boldsymbol{M}_n$. Lastly, our population risk for the value estimation takes the form

$$\mathcal{L}_n = \frac{1}{N} \mathrm{Tr} \boldsymbol{M}_n \bar{\boldsymbol{\Sigma}} + \frac{1}{T} \sum_t \sigma_t^2 \tag{B.25}$$

where $\frac{1}{T} \sum_t \sigma_t^2$ exactly quantifies the variance in rewards unexplained by state-dependent features.

### B.6 Tracking Iterate Moments with Direct Recurrence Relation

In this section we give a direct calculation of the first two moments of $\boldsymbol{w}$ over the collection of randomly sampled features $\{\psi_n^\mu(t)\}$ and show which terms are disregarded in the proportional limit examined in the main text.

Letting $\boldsymbol{A} = \bar{\boldsymbol{\Sigma}} - \gamma \bar{\boldsymbol{\Sigma}}_+$, we note that the average evolution of $\boldsymbol{w}$ has the form

$$\langle \boldsymbol{w}_{n+1} \rangle = (\boldsymbol{\Sigma} - \gamma \boldsymbol{\Sigma}_+) (\boldsymbol{w}_{TD} - \langle \boldsymbol{w}_n \rangle) \tag{B.26}$$

Thus, if we disregarded fluctuations in $\boldsymbol{w}_n$ due to SGD, the model will converge to the correct fixed point. Next, we look at $\boldsymbol{M}_n = \langle (\boldsymbol{w}_n - \boldsymbol{w}_{TD})(\boldsymbol{w}_n - \boldsymbol{w}_{TD}) \rangle$. Under the Gaussian equivalence ansatz, we have

$$\boldsymbol{M}_{n+1} = \boldsymbol{M}_n - \eta \boldsymbol{A} \boldsymbol{M}_n - \eta \boldsymbol{M}_n \boldsymbol{A}^\top + \frac{\eta^2}{T^2 B^2} \sum_{\mu\nu tt'} \langle \Delta_n^\mu(t) \Delta_n^\nu(t') \psi_n^\mu(t) \psi_n^\nu(t') \rangle$$

$$= (\boldsymbol{I} - \eta \boldsymbol{A}) \boldsymbol{M}_n (\boldsymbol{I} - \eta \boldsymbol{A})^\top - \frac{\eta^2}{B} \boldsymbol{A} \boldsymbol{M}_n \boldsymbol{A}^\top + \frac{\eta^2}{T^2 B} \sum_{tt'} \langle \Delta_n(t) \Delta_n(t') \psi(t) \psi(t')^\top \rangle$$

$$= (\boldsymbol{I} - \eta \boldsymbol{A}) \boldsymbol{M}_n (\boldsymbol{I} - \eta \boldsymbol{A})^\top + \frac{\eta^2}{T^2 B} \sum_{tt'} Q_n(t, t') \boldsymbol{\Sigma}(t, t')$$

$$+ \frac{\eta^2}{T^2 B} \sum_{tt'} \langle \Delta_n(t') \psi(t) \rangle \langle \Delta_n(t) \psi(t')^\top \rangle \tag{B.27}$$

S7

The mean field theory derived from saddle point integration consists of the first two terms in the final expression. Therefore mean field theory disregards the last term which computes cross time correlations of RPEs with features, effectively making the approximation

$$\frac{\eta^2}{T^2 B} \sum_{tt'} \langle \Delta_n(t')\boldsymbol{\psi}(t) \rangle \langle \Delta_n(t)\boldsymbol{\psi}(t')^\top \rangle \approx 0. \tag{B.28}$$

After making this approximation, we recover the learning curve obtained in the previous Section B.3. We show in our experiments that dropping this term does not significantly alter the learning curves.

## B.7  Scaling of Asymptotic Fixed Points

To identify fixed points in the value error dynamics, we can seek non-vanishing fixed points for the weight error covariance $\boldsymbol{M} = \langle (\boldsymbol{w} - \boldsymbol{w}_{TD})(\boldsymbol{w} - \boldsymbol{w}_{TD}) \rangle$. We note that $\langle \boldsymbol{w} \rangle \sim \boldsymbol{w}_{TD}$ asymptotically. Again, letting $\boldsymbol{A} = \bar{\boldsymbol{\Sigma}} - \gamma\bar{\boldsymbol{\Sigma}}_+$, we obtain the following fixed point condition for $\boldsymbol{M}$ under these assumptions

$$\boldsymbol{AM}+\boldsymbol{MA}^\top - \eta\boldsymbol{AMA}^\top = \frac{\eta}{BT^2}\sum_{tt'} Q(t,t')\boldsymbol{\Sigma}(t,t')$$

$$Q(t,t') = \mathrm{Tr}\boldsymbol{M}\boldsymbol{\Sigma}(t,t') - \gamma\mathrm{Tr}\boldsymbol{M}\left[\boldsymbol{\Sigma}(t,t'+1) + \boldsymbol{\Sigma}(t+1,t')\right] + \gamma^2\mathrm{Tr}\boldsymbol{M}\boldsymbol{\Sigma}(t+1,t'+1)$$
$$+ \gamma^2\boldsymbol{w}_{TD}^\top\bar{\boldsymbol{\Sigma}}^{-1}\bar{\boldsymbol{\Sigma}}_+\boldsymbol{\Sigma}(t,t')\bar{\boldsymbol{\Sigma}}_+\bar{\boldsymbol{\Sigma}}^{-1}\boldsymbol{w}_{TD} + \gamma^2\boldsymbol{w}_{TD}^\top\boldsymbol{\Sigma}(t+1,t'+1)\boldsymbol{w}_{TD}$$
$$+ \gamma^2\boldsymbol{w}_{TD}\bar{\boldsymbol{\Sigma}}^{-1}\bar{\boldsymbol{\Sigma}}_+\left[\boldsymbol{\Sigma}(t,t'+1) + \boldsymbol{\Sigma}(t+1,t')\right]\boldsymbol{w}_{TD}. \tag{B.29}$$

Where we used the formula for $Q_n(t,t')$ from Appendix B.6, evaluated at $\langle \boldsymbol{w} \rangle = \boldsymbol{w}_{TD}$ and used the fact that $\boldsymbol{w}_R = \boldsymbol{w}_{TD} - \gamma\bar{\boldsymbol{\Sigma}}^{-1}\bar{\boldsymbol{\Sigma}}_+\boldsymbol{w}_{TD}$. The solution $\boldsymbol{M} = 0$ is a valid fixed point for $\boldsymbol{M}$ in the $\eta \to 0$ and $B \to \infty$ limits because the constant terms on the right-hand side vanish. Similarly, if $\gamma = 0$ (which corresponds to the standard supervised learning case), the right hand side is linear in $\boldsymbol{M}$, allowing $\boldsymbol{M} = 0$ to be a valid fixed point.

However, for finite $B$ and non-zero $\eta$ and $\gamma$, there exists a solution to the above fixed point equation. For small $\frac{\eta\gamma^2}{B}$, we can deduce that $\boldsymbol{M}$ must satisfy a self-consistent asymptotic scaling of the form

$$\boldsymbol{M} = \mathcal{O}\left(\frac{\eta\gamma^2}{B}\right) \tag{B.30}$$

implying an asymptotic value error scaling of $\mathcal{L} \sim \mathrm{Tr}\boldsymbol{M}\bar{\boldsymbol{\Sigma}} \sim \mathcal{O}\left(\frac{\eta\gamma^2}{B}\right)$. These scalings are examined in Figure 3 where experiments obey the expected behavior.

## C  Reward Shaping

In this section, we consider the role of reward shaping on the dynamics of TD learning. As discussed in the main text, we consider potential based shaping with potential function decomposable in the features $\phi(s) = \boldsymbol{w}_\phi \cdot \boldsymbol{\psi}(s)$. We first describe the change to the average weight evolution $\langle \boldsymbol{w}_n \rangle$ and then describe the dynamics of the correlations. In potential based shaping, the TD errors take the form

$$\Delta(t) = R(s(t)) + \phi(s(t)) - \gamma\phi(s(t+1)) + \gamma\hat{V}(s(t+1)) - \hat{V}(s(t)) \tag{C.1}$$

Computing from the DMFT equations the evolution of $\langle \boldsymbol{w}_n \rangle$ we have

$$\langle \boldsymbol{w}_{n+1} \rangle = \langle \boldsymbol{w}_n \rangle + \eta\bar{\boldsymbol{\Sigma}}(\boldsymbol{w}_R + \boldsymbol{w}_\phi - \langle \boldsymbol{w}_n \rangle) + \gamma\eta\bar{\boldsymbol{\Sigma}}_+(\langle \boldsymbol{w}_n \rangle - \boldsymbol{w}_\phi)$$
$$= \langle \boldsymbol{w}_n \rangle - \eta\boldsymbol{A}\left[\boldsymbol{w}_{TD} + \boldsymbol{w}_\phi - \langle \boldsymbol{w}_n \rangle\right]. \tag{C.2}$$

We see that including the reward shaping function $\phi$ offsets the fixed point of the algorithm to be $\boldsymbol{w}_{TD} + \boldsymbol{w}_\phi$. This occurs precisely because the potential-based reward shaping generates an additive correction to the target value function by $\phi(s)$ [64]. When we predict value at evaluation, we use the reshifted value $\hat{V}(s) - \phi(s)$. The natural quantity to track at the level of the mean field equations is the adapted version of $\boldsymbol{M}_n$

$$\boldsymbol{M}_n = \left\langle (\boldsymbol{w}_n - \boldsymbol{w}_{TD} - \boldsymbol{w}_\phi)(\boldsymbol{w}_n - \boldsymbol{w}_{TD} - \boldsymbol{w}_\phi)^\top \right\rangle. \tag{C.3}$$

This correlation matrix has dynamics

$$M_{n+1} = (I - \eta A) M_n (I - \eta A)^\top + \frac{\eta^2}{BT^2} \sum_{tt'} Q_n(t,t') \Sigma(t,t') \tag{C.4}$$

and the TD-error correlations $Q_n(t,t')$ have the form

$$
\begin{aligned}
Q_n(t,t') &= \left\langle (w_R + w_\phi - w_n)^\top \Sigma(t,t')(w_R + w_\phi - w_n) \right\rangle \\
&+ \gamma \left\langle (w - w_\phi)^\top \left[ \Sigma(t,t') + \Sigma(t',t) \right] (w_R + w_\phi - w_n) \right\rangle \\
&+ \gamma^2 \left\langle (w_n - w_\phi)^\top \Sigma(t+1, t'+1)(w_n - w_\phi) \right\rangle
\end{aligned}
\tag{C.5}
$$

The value estimation error is again $\mathcal{L}_n = \mathrm{Tr} M_n \bar{\Sigma}$. We see that the two primary ways that reward shaping alters the loss dynamics is

- A change in the initial condition for $M_n$ to be $M_0 = (w_{TD} + w_\phi)(w_{TD} + w_\phi)^\top$

- A change in the TD error covariance term $Q_n(t,t')$

Both effects can generate significant changes in the dynamics and plateaus of the model.

# D   Non-Gaussian Features

The full non-asymptotic theory (no assumptions on $N, B$ large) of TD dynamics with linear function approximation closes under the fourth moments of the features. In this setting we do not incorporate explicit factors of $N$ in the defintion of the value estimator $\hat{V}(t) = w \cdot \psi(t)$. As before, we track the update to the $M$ matrix

$$
\begin{aligned}
M_{n+1} &= M_n - \eta A M_n - \eta M_n A^\top + \frac{\eta^2(B-1)}{B} A M_n A^\top \\
&+ \frac{\eta^2}{B} \sum_{tt'} \left\langle \Delta_n(t) \Delta_n(t') \psi(t) \psi(t')^\top \right\rangle
\end{aligned}
\tag{D.1}
$$

To calculate the last term, we introduce the following tensor of fourth moments

$$\kappa^4_{ijkl}(t_1, t_2, t_3, t_4) = \left\langle \psi_i(t_1)\psi_j(t_2)\psi_k(t_3)\psi_l(t_4) \right\rangle \tag{D.2}$$

In the Gaussian case, this reduces to an expression involving only the correlations. For example, if the features are zero mean, then we can use Wick's theorem to obtain the decomposition

$$\kappa^{4,Gauss}_{ijkl}(t_1, t_2, t_3, t_4) = \Sigma_{ij}(t_1, t_2)\Sigma_{kl}(t_3, t_4) + \Sigma_{ik}(t_1, t_3)\Sigma_{jl}(t_2, t_4) + \Sigma_{ij}(t_1, t_2)\Sigma_{kl}(t_3, t_4) \tag{D.3}$$

Now, using the fact that $\Delta_n(t) = (w_R - w_n) \cdot \psi(t) + \gamma w_n \cdot \psi(t+1)$, we find

$$
\begin{aligned}
&\left\langle \Delta_n(t) \Delta_n(t') \psi_i(t) \psi_j(t')^\top \right\rangle \\
&= \left\langle \psi_i(t)\psi_j(t') \, \psi(t)^\top (w_R - w_n)(w_R - w_n)^\top \psi(t') \right\rangle \\
&+ \gamma \left\langle \psi_i(t)\psi_j(t') \, \psi(t)^\top (w_R - w_n)w_n^\top \psi(t'+1) \right\rangle \\
&+ \gamma \left\langle \psi_i(t)\psi_j(t') \, \psi(t+1)^\top w_n (w_R - w_n)^\top \psi(t') \right\rangle \\
&+ \gamma^2 \left\langle \psi_i(t)\psi_j(t') \, \psi(t+1)^\top w_n w_n^\top \psi(t'+1) \right\rangle
\end{aligned}
\tag{D.4}
$$

Putting this all together, we find the following recurrence for $M_n$ in the non-Gaussian case

$$M_{n+1} = M_n - \eta A M_n - \eta M_n A^\top + \frac{\eta^2(B-1)}{B} A M_n A^\top$$

$$+ \frac{\eta^2}{BT^2} \sum_{t,t'} \mathrm{Tr} \kappa(t,t',t,t') \left\langle (w_R - w_n)(w_R - w_n)^\top \right\rangle$$

$$+ \frac{\gamma \eta^2 T^2}{B} \sum_{t,t'} \mathrm{Tr}\, \kappa(t,t',t,t'+1) \left\langle w_n(w_R - w_n)^\top \right\rangle$$

$$+ \frac{\gamma \eta^2}{BT^2} \sum_{t,t'} \mathrm{Tr}\, \kappa(t,t',t+1,t') \left\langle (w_R - w_n)w_n^\top \right\rangle$$

$$+ \frac{\gamma^2 \eta^2}{BT^2} \sum_{t,t'} \mathrm{Tr}\, \kappa(t,t',t+1,t'+1) \left\langle w_n w_n^\top \right\rangle \tag{D.5}$$

where all traces are taken against the last two time and feature indices. The averages over the weights close in terms of the average $\langle w_n \rangle$ and the covariance $M_n$. This equation is exact for any feature distribution, but requires significant computational resources to evaluate and is less illuminating than the mean field limit analyzed in the previous sections.

## D.1 Breakdown of Gaussian Theory in Low Dimension

In this section, we discuss the breakdown of Gaussian theory at low dimension $N$. In Figure D.1 we provide an example where the non-Gaussian distributions exhibit noticeably different learning curves than the Gaussian approximate theory (dashed black) and Gaussian samples with matching covariance (dashed color). We use the features

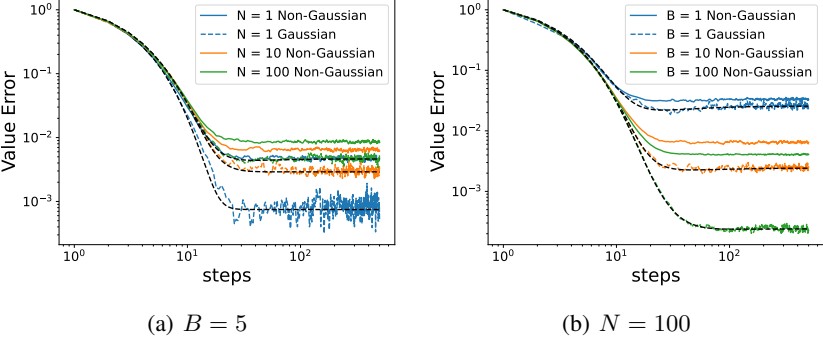

(a) $B = 5$            (b) $N = 100$

Figure D.1: The Gaussian theory can break down for non-Gaussian features in low dimension $N$. Illustration of the possible gap between Gaussian and non-Gaussian performance in the power law features of Figure 3 and defined in Appendix G. (a) The learning curves for batchsize $B = 5$ and varying dimension $N$. As $N$ increases the gap between the non-Gaussian experiment (solid) and the Gaussian theory (black dashed) decreases.

## D.2 A Simple Solveable Example

We next examine a very simple case where we can exactly characterize the gap between the non-Gaussian and Gaussian distributions. In this section, we examine the special case of $T = 1$ and look at features which are independent $p(\psi) = \prod_{i=1}^N p(\psi_i)$ (form a factor distribution). In this case, we obtain the following exact learning curve

$$\mathcal{L}_n = \left[ (1-\eta)^2 + \frac{\eta^2(N+1+\kappa)}{B} \right]^n \ , \ \kappa = \left\langle \psi^4 \right\rangle - 3 \left\langle \psi^2 \right\rangle^2 \tag{D.6}$$

which holds for any $N, B$. We note that in the limit where $N, B \to \infty$ with $B/N = \alpha$, we see that the dependence on $\kappa$ disappears and we arrive at the universal behavior

$$\mathcal{L}_n \sim \left[(1-\eta)^2 + \frac{\eta^2}{\alpha}\right]^n \quad, \quad N, B \to \infty \tag{D.7}$$

For example, we can consider vectors on the hypercube where $\psi_k \in \{\pm 1\}$ with equal probability for $k \in \{1, ..., N\}$ for the non-Gaussian distribution and compare to the Gaussian with identical covariance $\psi \sim \mathcal{N}(0, I)$.

$$\mathcal{L}_n = \begin{cases} \left[(1-\eta)^2 + \frac{\eta^2(N+1)}{B}\right]^n & \text{Gaussian } \psi \\ \left[(1-\eta)^2 + \frac{\eta^2(N-1)}{B}\right]^n & \text{Hypercube } \psi \end{cases} \tag{D.8}$$

The reason for the discrepancy between the Gaussian and Bernoulli/Hypercube loss curves is exactly the negative kurtosis of the hypercube features

$$\kappa_{\text{Gauss}} = 0$$

$$\kappa_{\text{Bernoulli}} - 3\Sigma^2_{\text{Bernoulli}} = \left\langle \psi^4 \right\rangle - 3 \left\langle \psi^2 \right\rangle^2 = -2 \tag{D.9}$$

An example of this result for low and high dimensions $N$ with $B = \alpha N$ with $\alpha = 0.1$ is provided in Figure D.2. In low dimension ($N = 10$) the Bernoulli/Hypercube feature have noticeably different dynamics than the Gaussian features. In the proportional limit $N, B \to \infty$ with $\alpha = B/N$ these learning curves are identical and all have the form $\mathcal{L}_n \sim \left[(1-\eta)^2 + \frac{\eta^2}{\alpha}\right]^n$.

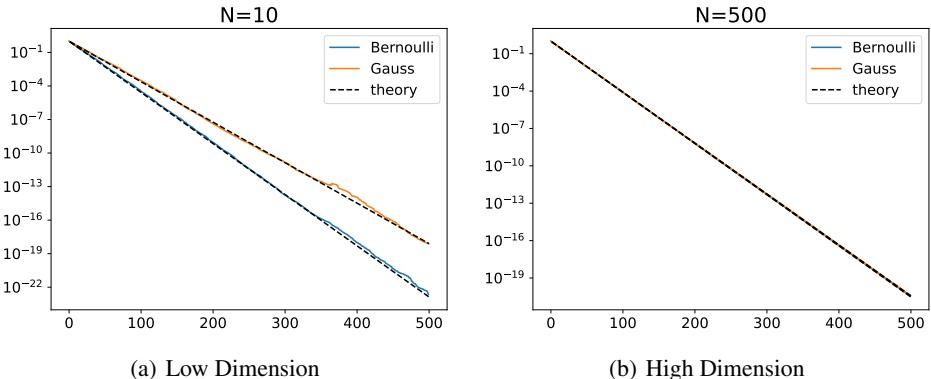

(a) Low Dimension            (b) High Dimension

Figure D.2: A simple explicitly solveable case shows how non-Gaussian corrections appear at finite size but disappear in the proportional limit where $N, B \to \infty$ with $\alpha = B/N = \mathcal{O}(1)$.

# E    Tests on Other Feature Distributions

In this section, we include additional tests of our theory on alternative features with the same random walk policy as in Figure 1.

# F    Plateau Scaling in MountainCar-v0 Environment

We verified the results of the theory on the environment MountainCar-v0. First, we train a policy with tabular $\epsilon$-greedy Q-Learning ($\epsilon = 0.1, \gamma = 0.99, \eta = 0.01$) to learn policy $\pi$. The position and velocity are discretized into 42 and 28 states, respectively. The learned policy $\pi$ is not optimal but consistently reaches goal within 350 timesteps. Therefore, each episode is set to have a length of 350 timesteps. Next, we take $\pi$ and evaluate it with TD learning.

Since MountainCar-v0 is a continuous environment, there is no closed solution to the ground truth of the value function. To estimate the ground truth value function, we ran TD learning with a small learning rate for 10M batches ($\eta = 0.01, B = 1, \gamma = 0.99$) to obtain $V^\pi \approx \hat{V}_{10M}$.

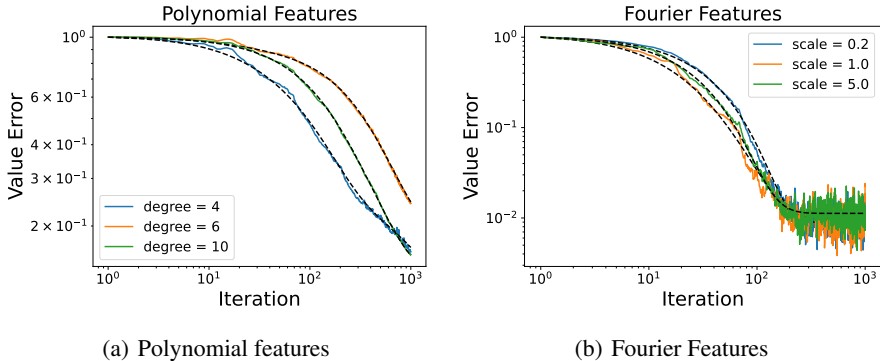

(a) Polynomial features          (b) Fourier Features

Figure E.1: In this Figure, we simulate the same random walk policy on the 2D grid world but use other types of features beyond RBF place cells. Learning dynamics are still accurately described by our theory. (c) The polynomial basis over states with random powers is constructed as $\psi_i(s_1, s_2) = s_1^{c_{i,1}} s_2^{c_{i,2}}$ where $c_{i,1}, c_{i,2}$ are chosen at random from $\{0, 1, ..., k\}$ where $k$ is the degree. (f) Fourier features with spectral power density $\frac{1}{1+\sigma^2(k_1^2+k_2^2)}$, where $\sigma$ is the scale/bandwidth.

# G   Numerical methods and additional details

The code to generate the Figures is provided in the Supplementary Material as a Jupyter Notebook at the following Github repository https://github.com/Pehlevan-Group/TD-RL-dynamics. Here, we briefly highlight some of the parameter choices.

For Figures 3 and 4 we use diagonally decoupled, but temporally correlated power law features with $\Sigma_{k\ell}(t, t') = \delta_{k\ell} \, k^{-1.2} \exp\left(-|t - t'|/\tau_k\right)$ with $\tau_k = \frac{10}{k+1}$ and $w_k^R = k^{-1.1}$ for $k \in [N]$ with $N = 300$. This type of feature structure is especially easy to evaluate the theoretical learning curves for. Unless otherwise stated, these figures used $\gamma = 0.9$ and batch size $B = 10$.

For the 2D MDP grid world, we defined a discrete set of states on a $17 \times 17$ grid. The agent starts in the middle position and follows a random diffusion policy where each possible movement (up, down, left, right) is taken with equal probability. The features were generated as bell-shaped place cells (shown). We computed $\Sigma(t, t')$ for the theory by sampling 5000 random draws of length $T = 50$. The Gaussian learning curve is obtained with TD learning with $\psi_G \sim \mathcal{N}(0, \Sigma)$.

Numerical experiments were performed on a NVIDIA SMX4-A100-80GB GPU using JAX to vectorize repetitive aspects of the experiments. With the exception of the MountainCar-v0 simulations, the numerical experiments (both preliminary experiments and those presented in the paper) took around 1 hour of compute time.

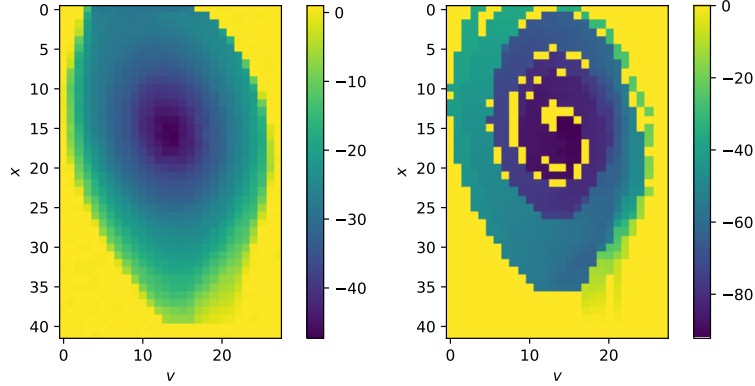

(a) $V^*$ estimated with tabular $\epsilon$-greedy Q-Learning

(b) $V^\pi$ TD Converged Value Function

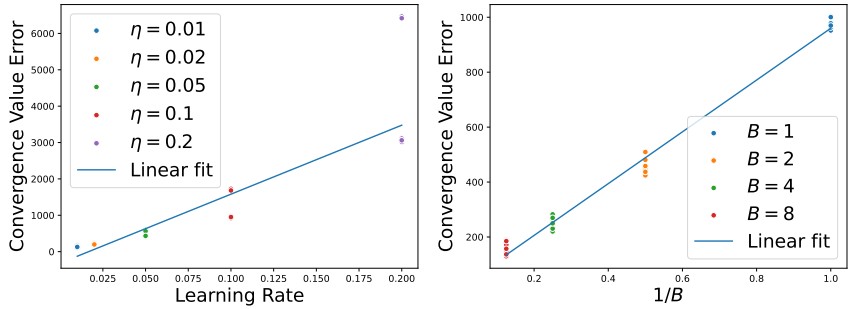

(c) Scaling of value error with learning rate

(d) Scaling of value error with batch size

Figure F.1: Simulation in a MountainCar-v0 environment. (a) Value function learned by Tabular Q-Learning that approximates the value function of an optimal policy. (b) An example value function of a policy ($V^\pi$) learned by TD learning. Notice that the value function does not equate to that in (a) due to the policy $\pi$ not reaching all states in the environment. (c-d) Linear scaling of convergence value error with the learning rate and the inverse of batch size. Target value function is the same across both experiments. Each dot represents a different seed. A total of 10 seeds were used. (c) Convergence value errors were computed by averaging the 100k batches before batch 10M. (d) Convergence value errors were computed by averaging the 100k batches before batch 1M.

