# OpenReview forum: "Loss Dynamics of Temporal Difference Reinforcement Learning"
_NeurIPS.cc/2023/Conference — NeurIPS 2023 poster_

### Official Review · Reviewer_mCei · 2023-06-08

**Soundness:** 4 excellent
**Presentation:** 4 excellent
**Contribution:** 4 excellent
**Rating:** 7
**Confidence:** 2

**Summary:**

In summary, they study how learning dynamics and plateaus depend on feature structure, learning rate, discount factor, and reward function in the case of batch TD(0) for policy evaluation.

This paper applies concepts from statistical physics to study typical case learning curves for TD learning in linear FAs with nonlinear but fixed features (specifically, their study is targeted at policy evaluation with batch TD(0)). They conjecture a Gaussian feature equivalence: roughly, having a high-dim feature vector, the learning curves of a TD learner are conjectured to be equivalent to that of one with Gaussian features having matching mean and correlations. Further, this view allows them to use a Spectral Perspective to find a proxy for what would constitute a harder reward function to optimize (showing alignment between learning rate progress vs. the measure; shaped reward being simpler than sparse reward).



**Strengths:**

- A step in an important direction: towards better understanding the learning dynamics of TD methods

- Proposing a creative tool for the analysis of TD methods

- Good exposition of related works

- Paper is well-motivated and clearly written

- Claims are carefully stated and supported: no overstated claims

- Experiments are pretty interesting and carefully designed

**Weaknesses:**

- Considering tasks have a finite horizon (fixed T), it has not been made clear if the experiments are run using features that have a good representation of the remaining time in them or not. Of course, this could be my misunderstanding (to be clarified by authors in the Questions section).

- Not sure whether the comparison between fixed and annealing learning rates make too much sense in the context of this study (to be clarified by authors in the Questions section).

**Questions:**

1. Are the results of the Discount Factor analysis (Fig. 3b) for a dense reward scenario? Wouldn’t this be a very different case with sparse rewards due to the issues studied by van Seijen et al. (2019)?

2. Isn’t annealing the learning rate a condition (even for tabular TD(0)) to guarantee asymptotic convergence to the optimal V with probability 1? If yes, what is the comparison of Fig. 5c vs 5d really showing us in the context of this paper?


- van Seijen *et al.* [NeurIPS 2019], "Using a Logarithmic Mapping to Enable Lower Discount Factors in Reinforcement Learning".


**Limitations:**

Limitations have been discussed sufficiently well.

---

> ### Author Rebuttal · Authors · 2023-08-08
>
> We thank the reviewer for the supportive review and good questions. We attempt to address the questions below.
>
> ### Response to Questions
>
> 1. About the discount factor analysis: The scenario we study is slightly different than the one studied by van Seijen et al. (2019). In our simualtions, we fix the reward function and compute the corresponding value function for each discount factor. We then compute the empirical and theoretical learning curves associated with each discount factor (which have different target value functions). We are therefore not studying the 'regularizing' effect of the discount factor, where there is a discrepancy between the metric used for performance and the discount factor used in hte learning rule, as investigated in van Seijen et al. (2019) or Amit. R., Meir, R. and Ciosek, K., 2020, November. Discount factor as a regularizer in reinforcement learning. ICML. Using our theory to understand this process would be an interesting avenue for future studies. In the experiments, the reward function was a sparse localized bump near the corner of the grid world.
> 2. About learning rate annealing: Learning rate annealing can be a condition to guarantee asymptotic convergence but here we examine a slightly different issue. The usual condition is that the annealing schedule for the learning rate $\eta_n$ at step n should satify: $\sum_{n=1}^{\infty} \eta_n =\infty$ and $\sum_{n=1}^{\infty} \eta_n^2 \lt  \infty$. But many possible annealing schedules satify these conditions and although they will eventually converge they will do so at different speeds. In Figure 3d-e, we show that our theory allows us to estimate the effect of the annealing schedule on the full learning dynamics and help choose a schedule that converges faster. In Figure 3c, there is no annealing and the dynamics reach a plateau which our theory can predict. In the additonal simulations in Figure 2 of the rebuttal PDF, we show that the predicted scaling of these plateaus is verified in a MountainCar environment. More generally, this point highlight how our approach differs from some theoretical work in RL which analyses bounds at convergence but does not provide a description of the learning dynamics.

---

> > ### Comment · Area_Chair_rjZu · 2023-08-17
> > **Reviewer response needed**
> >
> > Hello Reviewer,
> >
> > The authors have endeavoured to address your comments in their rebuttal. The rebuttal phase is a key part of the NeurIPS review process. I invite you to read and respond to the author's comments as soon as possible, latest tomorrow, to give everyone time to continue and conclude the discussion.
> >
> > Thank you for helping make NeurIPS a great conference for our community.

---

> > ### Comment · Reviewer_mCei · 2023-08-19
> > **Thanks for the clarifications**
> >
> > Thank you for your response. I remain positive about the paper and would like to see it accepted. My questions were clarified by the authors and as such I do not require any additions to the paper.

---

### Official Review · Reviewer_kcB9 · 2023-06-24

**Soundness:** 3 good
**Presentation:** 4 excellent
**Contribution:** 3 good
**Rating:** 7
**Confidence:** 4

**Summary:**

This paper looks to introduce a new theory for the learning dynamics in the online batch policy evaluation setting for temporal difference learning. The paper introduces the Gaussian Equivalence Conjecture, which postulates that the learning curves in TD can be modeled by Gaussian features (with per-time-step mean and covariance over features and all trajectories). Along with this modeling assumption, the work does not prove this conjecture, but instead assumes this conjecture to be true to show empirical results in modeling policy evaluation learning curves analytically on a simple gridworld MDP with “place cell” features.

Using this assumption and the example MDP, the work first shows results fitting their model with the learning curve of online policy evaluation, which is interesting in itself. Using this modeling assumption, the authors provide insight into the “hardness” of certain policy evaluation tasks and how it relates to their TD learning model with an example comparing sparse vs dense reward functions in policy evaluation. Beyond this, the work also shows examples (on the same domain) of their theory of learning dynamics predicting the effects of batch size, discount factor, learning rate and learning rate annealing. Finally, the work also uses this model to explain and predict what happens to learning dynamics in the case of reward shaping. The work uses this proposed model to reshape rewards to improve timescales of convergence by rotating the reward-scaling weights to be more aligned with features of high variance.

**Strengths:**

Overall, I think the work is of great interest to the reinforcement learning community with a good amount of both theoretical and empirical results to back their proposed model of learning dynamics. The paper poses and answers well-founded questions, and is also well written. I only have a few (albeit important) questions I’ll pose below.

**Weaknesses:**

My biggest issue with this paper in its current state is its set up. Currently, the proposed model is just a conjecture that seems to fit well with this one set of features (that are Gaussian it seems!) in this one gridworld environment. All the results seem to hinge on this conjecture being true, as evinced by the claim “We do not aim to provide a rigorous proof of this conjecture for TD learning but instead compute the learning curve implied by this assumption and compare to experiments on simple Markov Decision Processes”. Different forms of feature construction have been proposed throughout reinforcement learning history, including tile coding, polynomials, fourier basis etc. I’m still not convinced that this Gaussian model will carry over to learning dynamics with different features and different environments. While this is essentially my only concern with this paper (it’s very well written!), I think it is quite an important concern to address. Due to this concern, I have decided to only give this paper a weak accept. If more feature construction methods and more environments were included in these results, I would definitely be open to raising this score!

**Questions:**

Throughout reading this work, I only have a few section-by-section questions:

******3.1******

Could you give more specific details with regards to this environment? Maybe even in the appendix? I don’t exactly understand what you mean when you say ‘The feature map is parameterized by the bandwidth of individual “place cells”’.

******3.2******

For Remark 2 and 5, could you elaborate a bit on “fully explainable by features”? Is this referring to the case of partial observability? or is it just the most general formulation of R as a function of (s, a, s’)?

The notation for the proof for Proposition 3.1 is a bit confusing. I would suggest either simplifying by removing the notation that’s not defined here in the main paper and moving this to the appendix, or moving more of the appendix here.

**Limitations:**

Please see the general comment made in weaknesses to address limitations.

---

> ### Author Rebuttal · Authors · 2023-08-08
>
> We thank the reviewer for their support and useful questions and suggestions.
>
> ### Response to Weaknesses
> We thank the reviewer for their comment. These concerns were shared by others and we have added new theoretical results and simulations to show the generality of our approach.
> Specifcially, we have extended our theory to include a general formulation (see global response) in terms of fourth order moments of the features and show in Figure 1a-b of the rebuttal PDF that the gaussian model is a good approximation.
> In Figure 1c-d of the rebuttal PDF, we show that our theory also predicts the learning curves for agents with polynomial and fourier feature.
> In Figure 2 of the rebuttal PDF, we show that the scaling in terms of hyperparameters ($\eta, B$) predicted by our theory (equation B.24) is verified in a Mountain car environment
>
> ### Response to Questions
>
> 1. About Section 3.1: The 'place cell' terminology comes from neuroscience and refers to basis functions were each feature is a 2-D gaussian shape that is localized at a single location in the 2-D space (neurons with such receptive fields are found in the hippocampus). These are also called RBF features. In the simulations, we parametrize the feature maps by the width of the individual gaussian bumps. The environment is a 2-grid in which the agent performs an unbiased random walk. We will give an extended description of the environment and simulation details in the Appendix.
> 2. About Section 3.2: Note that we will merge this remarks into a single one. Here, we do not investigate the case of a partially observable MDP. Instead, 'fully explainable' refers to the case where the true value (or reward) function can be described with zero error within the space spanned by the features. If the true value (or reward) function has components that lie outside of the space spanned by the features, these components will never be explainable by a TD algorithms using this feature space. This is analagous to a target function outside the hypothesis class in supervised learning. In the Appendix, we provide the full formula taking into account the unexplainable case.
> 3. About the proof of Proposition 3.1: We will simplify our description of the proof outline in the main text. We will convey the intuition and methods used and we will link to the relevant equations in the Appendix instead.

---

> > ### Comment · Reviewer_kcB9 · 2023-08-14
> > **Response**
> >
> > Thank you for the additional results and responses.
> >
> > It seems that most other authors have the same quip with the work: the Gaussian Equivalence Conjecture. Given the new results presented in the rebuttal PDF, I believe the the Gaussian Equivalence Conjecture is a decent modelling assumption that represents a representative range of feature representations. With these new results, I am bumping my score up to a 7. If accepted, I believe the work to be a valuable contribution to the reinforcement learning community, and opens up new directions of research in trying to understand and model TD(0) learning curves.
> >
> > On that note, a final suggestion: instead of assuming a _conjecture_ over features, why not phrase it as a _modelling assumption_ instead? Much of reinforcement learning is just (good or bad) modelling assumptions made over the world. This change of argument might serve the paper better.

---

> > > ### Author Response · Authors · 2023-08-14
> > >
> > > We thank the reviewer for their appreciation of the new results we presented and the raising their score. The reviewer's suggestion is valuable, we will consider it while preparing the final submission. A related approach could be to first phrase the most general recursion which depends on fourth cumulants; and then state that our results simplify under a Gaussian features modeling assumption; and then state that the simplified form is more broadly applicable under the Gaussian Equivalence conjecture and present numerical evidence for it. We do not want to totally remove the Gaussian Equivalence conjecture, because we believe it should apply for high dimensional features. Further, it is much simpler to analyze than the most general recursion which depends on fourth moments (see rebuttal). Lastly, we think this approximation has the potential to trigger follow up work. We would also like to hear other reviewers' responses.

---

### Official Review · Reviewer_u1tm · 2023-07-06

**Soundness:** 2 fair
**Presentation:** 2 fair
**Contribution:** 1 poor
**Rating:** 3
**Confidence:** 3

**Summary:**

This work utilizes concepts from statistical physics to analyze the learning dynamics of TD(0) (policy evaluation) under the assumption of Gaussian equivalence, online batch update and linear function approximation. Using the theory, it demonstrates how learning rate annealing and reward shaping can improve learning in various aspects, based on multiple synthetic experiments.

**Strengths:**

The paper studies the learning dynamics of TD(0) from a novel perspective, which could provide readers with new insights and understandings

**Weaknesses:**

1. The whole framework relies on an important hypothesis that is not sufficiently verified in the TD(0) context. The Gaussian equivalence conjecture is only justified using a toy MDP (Fig.1) without explaining how accurate this hypothesis is for general MDPs. The provided example is using RBF features and RBF reward, which could be favorable to the given theory.

2. The connection to least-square TD methods is missing. In the same regime as the current paper, LSTD has been extensively studied for its convergence and learning dynamics (see, for example, Tagoti and Scherrer (2015) and Pan et al., (2017)). It does not rely on the Gaussian assumption and should be compared with the convergence result in the current paper.

Ref:
- Tagorti, M. and Scherrer, B., 2015, June. On the Rate of Convergence and Error Bounds for LSTD (λ). In *International Conference on Machine Learning* (pp. 1521-1529). PMLR.
- Pan, Y., White, A. and White, M., 2017, February. Accelerated gradient temporal difference learning. In *Proceedings of the AAAI Conference on Artificial Intelligence* (Vol. 31, No. 1).

3. It is unclear what Sec.4 is trying to demonstrate. The matrix A naturally appears in LSTD and has nothing special w.r.t. the current Gaussian equivalence setup.

Minor
- L136: correlation here means autocorrelation instead of Pearson product-moment correlation

**Questions:**

Q1: Any comments on why the Gaussian equivalence assumption would hold for general MDPs?

Q2: How are the current convergence results different from the typical results from the LSTD literature?

**Limitations:**

The paper did not sufficiently explain its connection to the literature.

---

> ### Author Rebuttal · Authors · 2023-08-08
>
>
> We thank the reviewer for their good questions and for pointing us to the LSTD literature. Below we address the weaknesses and questions.
> ### Response to Weaknesses
>
> We thank the reviewer for their comment and have added both derivations and simulations to show the generality of our framework. We analyzed non-RBF features including polynomial and fourier features (see Figure 1c-d of the rebuttal PDF). We also verify the predicted scaling of the fixed point of the dynamics with hyperparameters ($\eta, B$, equation B.24) for policy evaulation using TD(0) in a Mountain car environment (Figure 2 of rebuttal PDF)
>
> We also extended our theory to account for the non-asymptotic and non-Gaussian cases. Further, we also investigated more generally the question of Gaussian equivalence and looked at some simple exactly solveable models (at any finite $N,B$) where the joint limit is well described by Gaussian learning curve (is insensitive to higher moments in features, see Figure 1a-b of rebuttal PDF). There is a rich literature trying to establish universality in these high dimensional learning settings (see for instance Goldt et al, 2020, arXiv:2006.14709; Hu and Lu, 2020,arXiv:2009.07669; Gerace et al., 2022, arXiv:2205.13303).
>
> Second, we will add a detailed comparison with the LSTD literature to our related works section. In summary, we are merely studying online stochastic temporal difference learning where randomly sampled episodes (state sequences $\{ s_\mu(t) \}$) are used to estimate the gradient at each incremental step (more like SGD dynamics). This means at each iteration, the weights $w_n$ are updated incrementally with fresh samples of data. This is in contrast to LSTD algorithms where after observing $n$ samples, the best linear fit to the weights is performed (finding weights $w$ so that $\sum_{t=1}^n [\psi_t - \gamma \psi_{t+1}] \psi_t \cdot w = \sum_{t=1}^n \psi_t R_t$). In the papers cited by the reviewer, the authors are computing bounds on the error of the linear solution as a function of $n$. This generates a different kind of convergence rate (for example $O(n^{-1/2})$ in the first paper cited) than what we observe in the online/SGD setting, which are very dependent on feature structure and SGD noise. Further, the style of the analysis carried out in these papers (often worst case bounds) is different than what we pursue in the present work (average/typical case analysis in high dimension). Therefore, we believe our work presents a novel and complementary approach to study dynamics of convergence in reinforcement learning, specifically by studying the online setting of TD learning.
>
> Section 4 is trying to demonstrate what kinds of dynamics are observed when the SGD noise is negligible (The setting used in most LSTD papers and the approach taken by Lyle et al arXiv:2206.02126). This gives an ordering of reward/value functions which are easier or harder to learn. We agree that the $A$ matrix will also appear as the limiting value of the matrix in the LSTD algorithm and will comment on this similarity. This is a consequence of convergence of empirical covariances $\frac{1}{B} \sum_{\mu} \psi_\mu \psi_\mu^\top \to \Sigma$ to the population average of the feature covariance.
>
> We will specify that the correlations are not the pearson correlations in line 136.
>
> ### Response to Questions
>
> 1. We have worked out a general solution to the learning curves which closes at the level of the fourth cumulants (see global response). We also looked at some special cases where we can see why Gaussian equivalence should hold in high dimension. First, consider the case where each features $\psi_i$ is statistically independent. The variable $\hat{V}  = \psi \cdot w$ should obey a central limit theorem and behave as a Gaussian random variable when $N$ is large. Similarly, the random variable $\frac{1}{\sqrt B}\sum_{\mu=1}^B \Delta_\mu \psi_{\mu i}$ is also a sum of a large number of independent variables. We would thus expect the algorithm to depend only on the mean and variance of these variables. A concrete example where $T=1$ is provided in Figure 1 of the rebuttal PDF. We provide an exact solution and show that dependencies on higher cumulants vanish as $N,B\to\infty$.
> 2. The current convergence results are for online learning with SGD rather than for LSTD which can be considered as solving a full least squares problem at each iteration $n$. In the first case, the model does not perfectly fit the examples it has seen at finite $n$, whereas in LSTD, the model is fitting the observed samples as well as possible by solving the linear system
>     $A w = b \ , \ A = \sum_{t} (\phi_t - \gamma \phi_{t+1}) \phi_t^\top \ , \ b=\sum_t \phi_t R_t$
> Both algorithms are interesting as methods to learn a value function from samples, but the dynamics are both conceptually and practically distinct. We will clarify this in the paper.

---

> > ### Comment · Area_Chair_rjZu · 2023-08-17
> > **Reviewer response needed**
> >
> > Hello Reviewer,
> >
> > The authors have endeavoured to address your comments in their rebuttal. The rebuttal phase is a key part of the NeurIPS review process. I invite you to read and respond to the author's comments as soon as possible, latest tomorrow, to give everyone time to continue and conclude the discussion.
> >
> > Thank you for helping make NeurIPS a great conference for our community.

---

> > ### Comment · Reviewer_u1tm · 2023-08-21
> >
> > Thank you for the additional examples and results. However, the discussion regarding LSTD remains unsatisfactory. To begin with, LSTD can certainly be applied to incremental settings (Geramidfard et al., 2006) where samples are freshly generated at each step. More importantly, the current paper does not provide a clear and comparable result of convergence rates. For example, the annealing strategy gives the rate in L251, which is similar to Dalal et al. (2018, Theorem 3.1), and their results are not asymptotic. Finally, a similar analysis exists using Gaussian approximation (or CTL) for the convergence of LSTD (Konda 2002, Chapter 6).
> >
> > Overall, the paper did not discuss the existing literature in RL sufficiently, and the theoretical results are not significant enough. Thus I keep my score.
> >
> > Reference:
> > - Geramifard, A., Bowling, M., Zinkevich, M. and Sutton, R.S., 2006. iLSTD: Eligibility traces and convergence analysis. Advances in Neural Information Processing Systems, 19.
> > - Dalal, G., Szörényi, B., Thoppe, G. and Mannor, S., 2018, April. Finite sample analyses for TD (0) with function approximation. In Proceedings of the AAAI Conference on Artificial Intelligence (Vol. 32, No. 1).
> > - Konda, V., 2002. Actor-critic algorithms (Ph.D. thesis). Department of Electrical Engineering and Computer Science, Massachusetts Institute of Technology.

---

> > > ### Author Response · Authors · 2023-08-21
> > > **Response to Comment**
> > >
> > > We thank the reviewer for their response. We hope to clarify some of the issues raised above.
> > >
> > > 1. We are not claiming that LSTD cannot be used with incremental samples. We are stating that the **updates to the weights** in LSTD are not an instantaneous stochastic gradient update like in TD learning (what we study), but rather the instantaneous solution to a linear system of equations. Compare Section 2.1 and 2.2 of the paper [1]. For TD($\lambda$) (Section 2.1) the updates to weights are incremental given the new sample. In section 2.2 the new weights are solved as $w = A^{-1} b$.
> > > 2. We will cite Dalal et al about the upper bound they obtain for annealed learning rates. The bound in their Theorem 3.1 is distinct from our result (our typical case theory does not imply a scaling of $e^{- (\lambda / 2) n^{1-\sigma}}$ when annealing is sufficiently fast). However, their second term in their bound (when annealing slowly) would match the scaling of our derived asymptote $M \sim O(\eta_n) \sim O(n^{-\sigma})$. We will mention this.
> > > 3. The Konda et al argument uses the central limit theorem to estimate the error of the solution to the above linear system $w=A^{-1}b$. This is a standard tool in the analysis of linear systems where the matrix $A$ and vector $b$ are converging to steady state values. We instead are studying the dynamics of stochastic gradient TD updates through iterations. The vector $w_n$ has a different mean and covariance in TD learning compared to LSTD.
> > >
> > > As we mentioned in our rebuttal, we are planning on adding a related works section that describes the comparison of our work to LSTD and again explaining why it is different (like the difference between Section 2.1 and 2.2 in [1]).
> > >
> > > [1] Geramifard, A., Bowling, M., Zinkevich, M. and Sutton, R.S., 2006. iLSTD: Eligibility traces and convergence analysis. Advances in Neural Information Processing Systems, 19.

---

> ### Author Response · Authors · 2023-08-21
> **Review Discussion**
>
> We are following up to hear if our rebuttal addressed the main concerns of this reviewer or if they have any remaining questions that should be answered before the discussion period ends today. Any comments would be greatly appreciated. Thank you for your time.

---

### Official Review · Reviewer_KZaZ · 2023-07-08

**Soundness:** 3 good
**Presentation:** 2 fair
**Contribution:** 3 good
**Rating:** 5
**Confidence:** 3

**Summary:**

This paper provides a theoretical model that predicts the dynamics of TD learning. The theory assumes that the distribution of feature vectors is, in some sense, equivalent to a Gaussian distribution and predicts the value estimate at each iteration. The theory reveals a rich set of phenomena such as plateaus in TD learning.

**Strengths:**

The complete dynamics of TD learning have not been previously studied. And the paper provides a unique perspective on this problem.

The dynamics of TD learning according to the theoretical prediction are close to those in experiments, verifying the usefulness of the theory in these experiments.

The theory is also consistent with some existing tricks to improve TD learning such as reward shaping and step-size annealing.

**Weaknesses:**

The paper does not mention the limitations of its main assumption: the Gaussian Equivalence Conjecture. It verifies it using experiments in some small MDPs but these experiments do not tell when this assumption holds.

The paper does not explain clearly its theoretical results, making it difficult to understand these results.

The writing of the paper needs to be improved. Specifically, many sentences and notations remain confusing to readers. See Questions for details.

typos:

line 73: the features -> the vector of features
line 78: the offline setting
line 79: width -> with
line 245: two "learning"s

**Questions:**

Is T a constant or a random variable? If T is a constant you are considering the finite-horizon setting and the value function is horizon-dependent. If T is a random variable you need to specify how it is defined.

"the algorithms bootstraps its current predictions to estimate future states" Not sure what you mean here. I think you mean algorithm, not algorithms. And I think you mean bootstrapping from the estimated next state's value.

line 134: what are "the learning curves"? "high dimensional features" how many dimensions are needed? What do you mean when you say a set of learning curves is equivalent to another set of learning curves?

eqn 3: what is the meaning of <x>_{y} in your notation?

Line 138: "higher order cumulants of the features is negligible in high dimensional feature spaces under the square loss". Again, what do you mean by square loss. What do you mean by "higher" order "cumulants" of the features?

Figure 1: what is this diffusion process?

Proposition 3.1: what is the meaning of <>_s in the definition of L_n?

Line 191: this remark seems to be much more useful. Why not present your result without assuming that B -> infty? Could you briefly explain under what conditions the theory holds when B is not infinity?

Line 195: this remark also seems to be much more useful. Again, why not present your result without assuming that the value function is representable?

Line 203: how is this remark different from remark 2?

Line 205: what is w (without _n)?

Line 209: "which can be learned easily and which require more sampled trajectories" Why is it "easily" when it requires more sampled trajectories?

Line 216: "The theory predicts that," I do not see which part of your theory gives this result. Furthermore, how is this result different from the rate of convergence in a linear dynamical system?

Proposition 5.1: what is a fixed point of a learning curve?

How is eqn B.24 derived?

---

> ### Author Rebuttal · Authors · 2023-08-08
>
>
> We thank the reviewer for their careful reading of our paper and their detailed questions. We tried our best to clarify and address each of the weaknesses and questions raised below.
>
> **Response to Weaknesses**
>
> 1. We will add an acknowledgement that our paper relies on an assumption (Gaussian equivalence) which can not be validated or proven in general. Since the original submission, we have been able to derive more general error estimates (see Global response) and in some simple cases establish that Gaussian equivalence will hold in high dimension (learning curves don't depend on higher order cumulants, see Figure 1 of the rebuttal PDF). Based on this new result, we expect Gaussian equivalence under a variety of conditions including: (a) if the fourth moments are close to those of the corresponding Gaussian or (b) the features are close to independent (in some basis) and $N,B$ are both large. However, a more general set of conditions under which universality should hold would require a more careful proof so we will acknowledge this limitation explicitly in the discussion.
> 2. We thank the reviewer for pointing out areas where we can improve our writing and explanation of our results. We are adressing the points raised by the reviewer and listed below in Responses to Questions
>
> ### Responses to Questions
>
> We thank the reviewer for such detailed reading and questions. Below are our answers which we will include in the revised manuscript:
>
> 1. In our model $T$ is a constant that does not scale with the size of the features $N$ or the batch size $B$. We will state this more clearly in the problem setup.
> 2. We agree that we wrote this in a confusing way. We will fix this sentence so it reads *"the algorithm bootraps from the current estimate of the next state's value"*
> 3. line 134: by learning curve, we mean $\mathcal{L}_n$ the value estimation error as a function of iteration $n$. Our new theoretical work and simulations in Figure 1 of the rebuttal PDF, show that for high dimensional features, the contibution of the higher order cumulants is neglibible.
> 4. The notation $\left< x \right>_y$ is borrowed from physics and denotes an average of the function $x$ which depends on the random variable $y$.
> 5. Line 138: by the square loss, we mean that the evaluation metric is a square error between predicted and true value $(\hat{V} - V)^2$. By cumulant of features we mean fourth, fifth, etc. cumulants of the random variable $\psi_i(s_t)$. For example, the second cumulant is the covariance $\left< \psi_i(s_t) \psi_j(s_{t'}) \right> - \left< \psi_i(s_t) \right> \left<\psi_j(s_{t'}) \right>$. The fourth cumulant for mean zero random variables $\{ \psi_i \}$ is $\left< \psi_i \psi_j \psi_k \psi_l \right> - \left< \psi_i \psi_j \right>\left< \psi_k \psi_l \right> - \left< \psi_i \psi_k \right>\left< \psi_j \psi_l \right> -\left< \psi_i \psi_l \right>\left< \psi_j \psi_k \right>$. Generally, the cumulants are the coefficients in the Taylor series for the function $m(c) = \ln \mathbb{E}_{\psi} \exp\left( c \cdot \psi \right)$ around $c = 0$. Our new theoretical work shows we can handle arbitrary fourth cumulants $\kappa$ and our simulations show (Figure 1 of rebuttal PDF) that as assumed, the contribution of higher order cumulants becomes negligible as dimension increases
> 6. The diffusion process is an unbiased (isotropic) random walk in the 2D grid world state space.
> 7. $\left< \right>_s$ denotes average over states.
> 8. Line 191: We expect our result can hold at small $B$ if either the features are close to Gaussian (small fourth cumulant $\kappa$) or if the learning rate is small so that the SGD noise is small. In general, we expect the Gaussian equivalence to only kick in for large $N,B$, but we empirically observe that it can work well at small $N,B$ as well.
> 9. Line 195: We present the main text result with a representable value function to simplify the main text expression. We provide the full formula in the Appendix.
> 10. Thank you for pointing out the redundany in line 203. We will combine these two remarks.
> 11. Line 205: nice catch! It should be $w_n$. We will change this.
> 12. Line 209: This was a badly worded sentence. What we mean to say is that our theory can distinguish between easy and hard reward functions. We will reword the sentence appropriately.
> 13. Line 216: the theory predicts this expression since the equation for $M$ in the zero SGD noise limit is $M_{n+1} = (I-\eta A) M_n (I-\eta A)^\top$. Diagonalizing $A$ in its right eigenvectors, we can arrive at the spectral equation in line 216. We will add a detailed derivation in the appendix.
> 14. In proposition 5.1, the fixed point of the learning curve or the dynamics would correspond to a value where $L_{n+1}=L_{n}$ and $M_{n+1} = M_n$. We show that there can exist non-zero solutions to these equations that have a scale of $\mathcal{O}(\eta \gamma^2 /B)$.
> 15. There are many ways to derive equation B.24. Perhaps the simplest is to expand $M$ and $Q$ in a power series in $\eta$ so that $M = M_0 + \eta M_1 + \eta^2 M_2 + ...$ and $Q = Q_0 + \eta Q_1 + ...$. Plugging this into the fixed point condition in equation B.23 and solving order by order yields $M_0 = 0$ and $A M_1 + M_1 A = \frac{1}{T^2 B} \sum_{tt'} Q_0(t,t') \Sigma(t,t')$
> where $Q_0$ is independent of $M_1$.
> We see that the leading order scaling (in $\eta$) is thus $M \sim \mathcal{O}(\eta)$. This procedure can be repeated by expanding $M, Q$ in power series in $\gamma$ and $B^{-1}$ and extracting the leading order scalings, namely that $M\sim \mathcal{O}(\gamma^2)$ and $M \sim \mathcal{O}(B^{-1})$. Lastly, one can verify that $M \sim \mathcal{O}(\eta \gamma^2 /B )$ is self-consistent. We will add a more detailed set of arguments for this scaling in the Appendix.
>
> We hope these questions helped clarify our results. We will aim to improve the writing around these topics in the text and provide additional explanation of our notation and assumptions.

---

> > ### Author Response · Authors · 2023-08-19
> > **Discussion Follow Up**
> >
> > We are following up to hear if our rebuttal addressed the main concerns of this reviewer or if they have any remaining questions that should be answered before the discussion period ends. Thank you for your time.

---

### Official Review · Reviewer_5FZT · 2023-07-26

**Soundness:** 3 good
**Presentation:** 3 good
**Contribution:** 3 good
**Rating:** 5
**Confidence:** 1

**Summary:**

This paper introduces the concept of statistical physics to analyze reinforcement learning models. The authors propose a theory of learning dynamics for RL, with an emphasis on the role of linear function approximation. They investigate how strategies such as learning rate annealing and reward shaping can positively modify learning dynamics and surmount plateaus. The paper concludes by revealing new tools for developing a theory of learning dynamics in RL, thereby setting the stage for future research in this domain.

**Strengths:**

1. This paper introduces a novel method using statistical physics to formulate a theory of learning dynamics in RL.
2. The authors derive an analytical formula for the typical learning curve, demonstrating how their theory can predict the scaling of both learning convergence speed and performance plateaus based on problem parameters. This predictive ability is a significant advantage of the paper.
3. The paper outlines how this theory can assist in understanding and guiding design principles when selecting meta-parameters in RL algorithms. Such insights can enable practitioners to gain a deeper comprehension of how these factors influence the RL learning process.

**Weaknesses:**

1. The paper does have limitations regarding practical implications. Although it presents a theoretical framework and makes predictions about learning dynamics, the methods utilized, such as linear approximation, deviate significantly from current reinforcement learning (RL) methodologies.


**Questions:**

Is there a particular reason for choosing batch sizes that are not powers of 2, such as N=3, N=30, N=20, for the experiments? As far as I am aware, batch sizes that are powers of 2 are most commonly used for practical considerations.




**Limitations:**

No concerns about negative societal impact.

---

> ### Author Rebuttal · Authors · 2023-08-08
>
> We thank the reviewer for their comments and useful questions. We respond to the weaknesses and questions below.
>
> **Responses to Weaknesses**
>
> The reviewer is correct that our theory is limited to linear function approximation. This allows us to make strong predictions, but does not capture interesting aspects of deep learning such as feature learning. We address this in our limitations section but will expand on it further.
>
> However, we think that even in the linear model, some interesting statistical and dynamical phenomena such as SGD effects arise that are worthy of study (Paquette et al 2022 arXiv:2205.07069, Simon et al arXiv:2303.15438 2023, Mignacco et al 2020 arXiv:2006.06098).
>
> Further, if the Gaussian equivalence/universality ideas hold in RL, it could allow for future analyses of two layer neural neural networks which learn features as they optimize the value function (such as in Goldt et al arXiv:2006.14709 2020).
>
> **Response to Questions**
>
> There is no particular reason we chose the batch sizes we did. We can add a plot with powers of two if the reviewer thinks this is important. It would look very similar with plateaus evenly spaced on the log-scaled y-axis (see also scaling as a function of batch size on the MountainCar simulation in Figure 2 of rebuttal PDF).

---

> > ### Comment · Area_Chair_rjZu · 2023-08-17
> > **Reviewer response needed**
> >
> > Hello Reviewer,
> >
> > The authors have endeavoured to address your comments in their rebuttal. The rebuttal phase is a key part of the NeurIPS review process. I invite you to read and respond to the author's comments as soon as possible, latest tomorrow, to give everyone time to continue and conclude the discussion.
> >
> > Thank you for helping make NeurIPS a great conference for our community.

---

> > ### Comment · Reviewer_5FZT · 2023-08-19
> >
> > Thank you for the additional responses.
> >
> > I acknowledge the potential for extending the linear function approximation into future studies.
> > Regarding the batch sizes, I wanted to confirm that the authors haven't selectively chosen specific batch sizes that could unduly favor their hypothesis. I comprehend that the results have exhibited consistency across various batch sizes, and that the numbers have not been selectively chosen. As a suggestion, opting for numbers that are powers of two could help demonstrate that the choices are not biased.

---

> > > ### Author Response · Authors · 2023-08-19
> > > **Discussion Response**
> > >
> > > Thank you for your response!
> > >
> > > *Regarding the batch sizes, I wanted to confirm that the authors haven't selectively chosen specific batch sizes that could unduly favor their hypothesis. As a suggestion, opting for numbers that are powers of two could help demonstrate that the choices are not biased.*
> > >
> > > Yes, we confirm that these batch size choices are not biased. If the paper is accepted we will use powers of 2 for the batches in this figure. The plot still shows good agreement between experiment and theory.

---

> ### Author Response · Authors · 2023-08-19
> **Batchsize Powers of Two Data**
>
> Though we cannot add new links to figures at this point due to the response period policy, below we provide a table of the experimental and theoretical value errors $\mathcal{L}_n$ at times $n \in [1,10,100]$. The experimental losses are averaged over 10 random training experimetns. The full loss curve $\mathcal{L}_n$ will be included in the final version of the paper.
>
> batch = 1
> expt
> [0.62140151 0.04056274 0.03076534]
> theory
> [0.62642096 0.03724296 0.02816423]
>
> batch = 2
> expt
> [0.61854012 0.02638373 0.01317829]
> theory
> [0.62018211 0.0263546  0.01381489]
>
> batch = 4
> expt
> [0.58340318 0.02736459 0.00678785]
> theory
> [0.61706269 0.02101523 0.00684711]
>
> batch = 8
> expt
> [0.6271198  0.01924012 0.00419846]
> theory
> [0.61550298 0.01837126 0.00341322]
>
> batch = 16
> expt
> [0.62514669 0.01700518 0.00193409]
> theory
> [0.61472312 0.01705564 0.00170856]
>
> batch = 32
> expt
> [0.62464086 0.01665385 0.00081263]
> theory
> [0.61433319 0.01639942 0.00085928]

---

### Author Rebuttal · Authors · 2023-08-08

We thank the reviewers for their comments and suggestions for additional theoretical justifications and experiments. Based on the reviews, we have added a more in depth analysis of non-Gaussian features at arbitrary dimension and can show that the learning curves close under fourth moments of the features and that the predicted plateau of $\mathcal{O}(\frac{\eta \gamma^2}{B})$ is preserved. We have added more experiments testing the range of validity of our theory's predictions and figures which we will include in the revised manuscript (see the attached PDF) which show:
1. Simulations showing why the Gaussian equivalence ansatz is reasonable in the high dimension limit based on a toy model where $T=1$ (SGD) and each feature is independent. We solve the model for arbitrary fourth cumulant $\kappa$ at any dimension $N$ and find that the learning curve scales as $\left[(1-\eta)^2 + \frac{\eta^2(N + 1+\kappa)}{B} \right]^n$, which loses dependence on $\kappa$ in the large $N,B$ limit. This is intuitive since $\hat{V} = \frac{1}{\sqrt N} \sum_i \psi_i w_i$ should behave as a Gaussian random variable during training even if $\psi_i$ is non-Gaussian.
2. More experiments showing the accuracy of our theory on the random walk MDP with polynomial features of varying degrees and fourier features with different lengthscales/bandwidths. This provides more evidence that our theory (and the Gaussian approximation) is reasonable for many types of features.
3. We have run policy evaluation using TD(0) with a pre-trained policy on Mountain-Car ( we perform policy evaluation with Fourier features) and show that the loss curves exhibit asymptotes that scale linearly with $\eta$ and inversely with $B$. Since the feature space dimension $N$ and episode length $T$ are both very large, we do not attempt to estimate the $\Sigma(t,t')$ matrix. However, our theory's prediction of the asymptote allows us to accurately capture the scaling of the plateau with various hyperparameters.

We hope that these simulations give more evidence that our setting and assumptions are reasonable. Given more time, we can add additional tests of other types of features.


In addition, we also will make several changes to the paper
1. In the new version of our paper's Appendix we will also provide a derivation of a theoretical learning curve which is non-asymptotic (does not require $N,B$ large) and for any distribution. The SGD noise term in the resulting theory depends on the fourth moments of features $\kappa_{ijkl}(t_1,t_2,t_3,t_4) = \left< \psi_i(t_1)\psi_j(t_2)\psi_k(t_3)\psi_l(t_4) \right>$. The loss is still $\mathcal{L}_n = \frac{1}{N} \text{Tr} M_n \bar{\Sigma}$ but with the dynamics

\begin{align}
M_{n+1} &= M_n - \eta A M_n - \eta M_n A^\top + \frac{\eta^2 (B - 1)}{B} A M_n A^\top \nonumber
    \\
    &+ \frac{\eta^2}{B T^2} \sum_{t,t'} \text{Tr} \kappa(t,t',t,t') \left< (w_R -w_n)(w_R-w_n)^\top \right> \nonumber
    \\
    &+ \frac{\gamma \eta^2}{B  T^2}\sum_{t,t'} \text{Tr} \\ \kappa(t,t',t,t'+1) \left< w_n (w_R -w_n)^\top \right> \nonumber
    \\
    &+ \frac{\gamma \eta^2}{B T^2}\sum_{t,t'} \text{Tr} \\  \kappa(t,t',t+1,t') \left<  (w_R -w_n) w_n^\top \right> \nonumber
    \\
    &+ \frac{\gamma^2 \eta^2}{B T^2} \sum_{t,t'} \text{Tr} \\ \kappa(t,t',t+1,t'+1) \left<  w_n w_n^\top \right>
\end{align}
The trace is taken over the last two of the feature indices of $\kappa$. All weight averages can still be expressed in terms of $M_n$ like in the current Appendix. This prediction still recovers a potential fixed point in the dynamics at scale $\mathcal{O}(\eta\gamma^2/B)$, so this prediction is actually universal. Computing this theory is even less tractable than our original result due to the number of entries associated with the $\kappa$ object $\sim N^4 T^4$.

2. We also will provide more room to explain and introduce our notation (for instance that $\left< \right>$ denotes averaging) and will more carefully introduce the mathematical objects which appear in our main text.
3. We will provide a more extensive comparison of the algorithm considered in our work (online TD learning) to other algorithms in the literature including TD Least Squares (TDLS).

We hope that the reviewers will take these updates into consideration when re-evaluating our work.

---

### Decision · Program_Chairs · 2023-09-21

**Decision:**

Accept (poster)

**Comment:**

The paper uses statistical physics methods to analyse temporal difference learning in simple settings.

Strengths:

-Derives analytical results for typical learning curves in RL

-Predictions agree very well with experiments

-Tackles an important question and provides complementary insights to existing theoretical analyses in RL

-The results are clearly presented

Weaknesses:

-Relies on unproven assumptions (as with many statistical physics methods)

-Can only treat simple settings at present, and the broader applicability remains unclear